# Effects of hypoxia and non-lethal shell damage on shell mechanical and geochemical properties of a calcifying polychaete

Jonathan Y.S. Leung[1,2], Napo K.M. Cheung[2,3]

[1]School of Biological Sciences, The University of Adelaide, Adelaide, Australia

[2]Department of Biology and Chemistry, City University of Hong Kong, Hong Kong SAR

[3]Department of Biological Sciences, Graduate School of Science, The University of Tokyo, Tokyo, Japan

*Correspondence to*: Jonathan Y.S. Leung (jonathan_0919@hotmail.com)

**Abstract.** Calcification is a vital biomineralization process where calcifying organisms construct their calcareous
shells for protection. While this process is expected to deteriorate under hypoxia which reduces the metabolic energy yielded by aerobic respiration, some calcifying organisms were shown to maintain normal shell growth. The underlying mechanism remains largely unknown, but may be related to changing shell mineralogical properties, whereby shell growth is sustained at the expense of shell quality. Thus, we examined whether such plastic response is exhibited to alleviate the impact of hypoxia on calcification by assessing the shell growth and shell properties of a
calcifying polychaete in two contexts (life-threatening and unthreatened conditions). Although hypoxia substantially reduced respiration rate (i.e. less metabolic energy produced), shell growth was only slightly hindered without weakening mechanical strength under unthreatened conditions. Unexpectedly, hypoxia did not undermine defence response (i.e. enhanced shell growth and mechanical strength) under life-threatening conditions, which may be attributed to the changes in mineralogical properties (e.g. increased calcite/aragonite) to reduce the energy demand for
calcification. While more soluble shells (e.g. increased Mg/Ca in calcite) were produced under hypoxia as the trade-off, our findings suggest that mineralogical plasticity could be fundamental for calcifying organisms to maintain calcification under metabolic stress conditions.

## 1 Introduction

Calcification is a biomineralization process where many marine organisms, such as corals, molluscs, polychaetes and echinoderms, deposit carbonate minerals and form their calcareous shells or skeletons. This process is highly associated with the fitness and survival of calcifying organisms because shell growth not only allows continuous somatic growth, but also strengthens protection against physical and chemical damages. The protective role of shells is particularly important under life-threatening conditions (e.g. following non-lethal shell damage),
where many calcifying organisms are able to produce stronger shells at a higher rate to increase physical protection (Cheung et al., 2004; Brookes and Rochette, 2007; Hirsch et al., 2014). Indeed, such inducible defence response via enhanced calcification plays an important role in the survival of calcifying organisms (Harvell, 1990).

In view of the accelerated anthropogenic emission of carbon dioxide, calcification and hence defence response of calcifying organisms may be dampened by climate change stressors, such as ocean acidification and hypoxia (Bijma et al., 2013). While ocean acidification was expected to retard calcification (Orr et al., 2005), it is now realized that calcification is not primarily driven by the pH and carbonate saturation state of seawater (Roleda et al., 2012), meaning that the impact of ocean acidification on calcifying organisms through the changes in seawater carbonate chemistry is less deleterious than previously thought (e.g. Garilli et al., 2015; Ramajo et al., 2016; Leung et al., 2017a,b). Indeed, calcification is an energy-dependent physiological process actively regulated by calcifying organisms (Roleda et al., 2012); therefore, this process is likely determined by the energetics of calcifying organisms. As such, hypoxia (i.e. dissolved oxygen concentration in seawater $\leq 2.8$ mg $O_2$ $L^{-1}$ or $\leq 63$ µmol $L^{-1}$, Wu, 2002) can probably compromise calcification through its direct, adverse effect on aerobic metabolism and hence production of metabolic energy (Wu, 2002; Leung et al., 2013a). Since calcification is an energy-demanding process (Palmer, 1992), the impaired aerobic metabolism under hypoxia could be the underlying mechanism causing the reduced calcification as previously observed (e.g. Cheung et al., 2008; Findlay et al., 2009; Wijgerde et al., 2014). As the occurrence of hypoxia is predicted to become more prevalent in future marine ecosystems owing to ocean warming and human-induced eutrophication (Diaz and Rosenberg, 2008; Keeling et al., 2010; Bijma et al., 2013), the impact of hypoxia on calcifying organisms would be continuously escalated.

However, few previous studies showed that some calcifying organisms are able to maintain calcification under hypoxia (Mukherjee et al., 2013; Frieder et al., 2014; Keppel et al., 2016), and even anoxia (Nardelli et al., 2014). These unexpected results suggest potential mechanisms which can help compensate for the reduced metabolic energy under hypoxia in order to sustain calcification. This could be mediated by phenotypic plasticity, which involves trade-offs between phenotypic traits in response to altered conditions (Malausa et al., 2005). For example, shell growth may be maintained under hypoxia at the expense of shell quality or other physiological processes (e.g. soft tissue growth, reproduction and somatic maintenance) via energy trade-offs (Nisbet et al., 2012; Sokolova et al., 2012). Alternatively, energy demand for calcification may be reduced by changing geochemical properties of shells and thus favours shell growth when metabolic energy is reduced (Ramajo et al., 2015; Leung et al., 2017a). For instance, bimineralic calcifying organisms (i.e. organisms which can produce both calcite and aragonite) may precipitate a greater proportion of calcite to promote shell growth under metabolic stress conditions (e.g. ocean acidification, Chan et al., 2012; Leung et al., 2017a) because calcite has a lower packing density and its production requires less metabolic energy than aragonite (Weiner and Addadi, 1997; Hautmann, 2006). For calcite-producing organisms, a small quantity of magnesium ions is incorporated into the calcite lattice and impacts the quality of shells (e.g. solubility). While Mg incorporation could be physiologically regulated by calcifying organisms *per se* (Bentov and Erez, 2006), this energy-consuming regulation may be reduced under hypoxia so that more energy can be allocated to shell growth. To form crystalline calcium carbonate, metabolic energy is required for stabilization of amorphous calcium carbonate (ACC) because it involves some matrix proteins and transport of carbonate ions (Addadi et al., 2006; Bentov, 2010; Weiner and Addadi, 2011). In order to conserve energy for shell growth, therefore, less crystalline shells may be produced under hypoxia as the trade-off. Whether calcifying organisms can exhibit these plastic responses to alleviate the impact

of hypoxia-induced metabolic depression on calcification and defence response remains largely unknown and deserves a comprehensive investigation.

In this study, we examined how hypoxia affects calcification and defence response of a common calcifying polychaete (*Hydroides diramphus*), which is tolerant to hypoxia (Vaquer-Sunyer and Duarte, 2008; Leung et al., 2013b). Calcification was indicated by shell growth, while defence response by both shell growth and fracture toughness. We analysed the mineralogical properties of shells (organic matter content, calcite to aragonite ratio, magnesium to calcium ratio in calcite and relative amorphous calcium carbonate content) to indicate the possible changes in calcifying mechanism in response to hypoxia. Respiration rate and feeding rate were measured to reflect aerobic metabolism and energy gain, respectively. Given the possible impact of hypoxia on aerobic metabolism, we hypothesized that (1) the mineralogical properties of newly-produced shells would be modified to reduce the energy demand for calcification so that shell growth can be sustained; (2) defence response would be undermined as the reduced metabolic energy is possibly insufficient to enhance both shell growth and fracture toughness. If changing mineralogical properties of shells can help alleviate the impact of hypoxia on calcification and even defence response without causing significant adverse effects by trade-offs, this suggests that some calcifying organisms would be more robust to metabolic stress conditions than previously thought.

## 2 Materials and methods

### 2.1 Collection and maintenance of specimens

A calcifying polychaete *Hydroides diramphus* was selected as the study species, which lives on hard substrate and is widely distributed within circumtropical regions (Çinar, 2006). Adult polychaetes (tube length: 35 – 45 mm) were collected from a fish farm at Yung Shue O (22°25′N, 114°16′E), Hong Kong, in summer when hypoxia was commonly observed (Leung et al., 2013a). Other fouling organisms on the calcareous tube of *H. diramphus*, such as mussels and tunicates, were carefully removed. Then, the polychaetes were temporarily reared in plastic tanks (50 cm × 40 cm × 30 cm) filled with natural seawater under laboratory conditions (dissolved oxygen concentration: $6.00 \pm 0.10$ mg $O_2$ $L^{-1}$, pH: $8.10 \pm 0.05$, temperature: $28.0 \pm 1.0$°C and salinity: $33.0 \pm 0.5$ psu). Algal suspension containing live *Isochrysis galbana* and *Dunaliella tertiolecta* (1:1, v/v) was daily provided as food. The polychaetes were allowed to acclimate under these laboratory conditions for one week before experimentation.

### 2.2 Experimental design and rearing method

The impact of hypoxia on the calcification and defence response of adult *H. diramphus* was examined using a full factorial experimental design, involving two dissolved oxygen levels (normoxia vs. hypoxia) and two contexts (unthreatened vs. threatened). Thus, there were four treatment conditions based on their crossed combinations: (1) normoxia and unthreatened, (2) normoxia and threatened, (3) hypoxia and unthreatened, and (4) hypoxia and threatened. Normoxia (~6.0 mg $O_2$ $L^{-1}$, i.e. control) and hypoxia (~2.0 mg $O_2$ $L^{-1}$) were achieved by continuously

aerating seawater with air and a mixture of nitrogen and air, respectively (Leung et al., 2013b). Digital flow meters (Vögtlin Instruments, Switzerland) were used to adjust the flow rate of each gas (i.e. nitrogen and air) so that the desired dissolved oxygen concentration for hypoxia was maintained. To induce life-threatening condition for the polychaetes, non-lethal shell damage was made by carefully trimming the calcareous tube until the radioles were exposed, while the body was still fully covered. The polychaetes with "intact" (tube length: ~40 mm; body length: ~20 mm) and "damaged" (tube length: ~20 mm; body length: ~20 mm) tubes were then allowed to acclimate under either normoxia or hypoxia for another week before experimentation, which can particularly help the "damaged" polychaetes to recover from the stress induced by tube trimming (i.e. fight-or-flight response) so that they were only subject to the stress induced by non-lethal shell damage in the following experiments.

A total of 120 adult polychaetes were evenly and randomly assigned to each of the four treatment conditions (i.e. $n = 30$ polychaetes per treatment). The rearing method for the polychaetes was previously described (Leung and Cheung, 2017). Briefly, polychaetes with their initial tube length measured (see the section below) were individually transferred into 2-mL labelled microcentrifuge tubes with the radioles pointing upward. A small hole (~2 mm) was drilled at the bottom of each microcentrifuge tube to allow water exchange. The microcentrifuge tubes were glued together by hot-melt adhesives (3M, USA) to maintain an upright position and put into a lidded glass bottle (10 polychaetes per bottle; 3 bottles per treatment) containing 180 mL filtered seawater (FSW) (pore size: 0.45 μm). Bottles assigned to the same dissolved oxygen level (i.e. normoxia or hypoxia) were connected to the same gas inlet and had the target dissolved oxygen concentration manipulated as described above. Stable equilibrium between gases in seawater was achieved rapidly by this constant aeration (< 5 min) and thus the target dissolved oxygen concentration in seawater, which was daily recorded using an optical dissolved oxygen probe (SOO-100, TauTheta Instruments, USA), was very stable over time (Fig. A1). To simulate the summer seawater temperature at the collection site, the whole setup was incubated in a water bath with temperature maintained at 28°C using a heating bath circulator. The polychaetes were reared under a day/light cycle of 14:10 h. Algal suspension (20 mL) containing live *I. galbana* and *D. tertiolecta* (1:1, v/v) at ~$1 \times 10^6$ cells mL$^{-1}$ was provided daily as food to ensure adequate food supply for normal shell growth. The microcentrifuge tubes were cleaned and the seawater was gently renewed once every three days to prevent accumulation of excreted waste. The exposure lasted for 3 weeks, excluding the initial acclimation period. After the 3-week exposure period, only 4 out of 120 polychaetes died across treatments (2 from "Intact, Normoxia" and 2 from "Damaged, Hypoxia"), meaning that the treatment conditions *per se* did not cause fatality.

### 2.3 Shell growth

Shell growth was indicated by the increase in tube length over time, where the newly-produced shells can be easily identified by the difference in colour from the original shells (Fig. 1). The tube length of all individuals was measured on Day 1, Day 11 and Day 21 to estimate shell growth ($n = 30$ polychaetes per treatment). During the tube length measurement, a polychaete was temporarily placed in a Petri dish (diameter: 90 mm) filled with seawater at their respective dissolved oxygen concentration to avoid potential desiccation. Tube length was measured under a dissecting microscope with a scale to the nearest 0.1 mm, followed by putting the polychaete back to the respective

glass bottle immediately (< 30 s for each measurement). Since tube growth can be measured with sufficient accuracy and precision under the dissecting microscope, the tube growth of each individual was analysed as a replicate.

**2.4 Physiological performance**

Following the 3-week exposure period, the respiration rate and feeding rate of polychaetes were measured using the method described in Leung et al. (2013a) with minor modifications. Briefly, 25 individuals from the same treatment were randomly sampled and evenly transferred into five airtight polypropylene syringes (Terumo® hypodermic syringe without needle, Terumo Corporation, Japan) each containing ~35 mL FSW with dissolved oxygen concentration adjusted to the corresponding treatment level ($n = 5$ replicate syringes per treatment). They were allowed to rest in the syringe for 15 min. Then, the initial dissolved oxygen concentration of FSW was measured using an optical dissolved oxygen probe (SOO-100, TauTheta Instruments, USA), calibrated according to the manual of manufacturer. The atmospheric air inside the syringe, which helps buffer the change in dissolved oxygen concentration during the resting period, was then fully expelled and the tip of the syringe was sealed by Blu Tack to ensure an airtight condition. After one hour, the final dissolved oxygen concentration of FSW was recorded when it becomes steady by gently stirring the FSW to ensure uniform dissolved oxygen concentration inside the syringe. Blank samples without individuals were prepared to correct the background change in dissolved oxygen concentration, which fluctuated less than 1%. Respiration rate was expressed as $\mu g$ $O_2$ $ind^{-1}$ $hr^{-1}$.

To measure feeding rate, we determined the decrease in concentration of microalgae in a given period of time (i.e. clearance rate), as previously described (Riisgård, 2001; Contreras et al., 2012; Leung et al., 2013a; Leung and Cheung, 2017). For each treatment, 25 randomly selected individuals, which had been starved for one day to standardize their hunger level, were put into five glass vials (i.e. $n = 5$ replicate glass vials per treatment) each containing 80 mL FSW with an initial concentration of ~$1 \times 10^6$ cell $mL^{-1}$ live *D. tertiolecta*. After feeding for one hour under light conditions, 1 mL seawater was taken from the bottle and the microalgae were enumerated using a haemocytometer (6 trials per bottle). Prior to counting, 1% Lugol's solution was used to fix the microalgae. Clearance rate was calculated using the following formula to represent feeding rate (Coughlan, 1969):

$$CR = \frac{V}{nt} \times ln\frac{C_o}{C_t}$$

where $CR$ is the clearance rate (mL $ind^{-1}$ $hr^{-1}$); $V$ is the volume of seawater; $n$ is the number of individuals; $t$ is the feeding time; $C_o$ and $C_t$ are the initial and final concentrations of microalgae, respectively.

**2.5 Shell properties**

After measuring respiration rate and feeding rate, the newly-produced shells for the analyses of mechanical
and geochemical properties were carefully removed using a pair of forceps and then rinsed with deionized water to
remove the microalgae and other debris on the shell surface.

Fracture toughness was measured using a micro-hardness tester (Fischerscope HM2000, Fischer, Germany)
to indicate mechanical strength. For each treatment, five shell fragments from five randomly selected individuals were
mounted firmly onto a metal disc with the inner shell surface facing upwards using cyanoacrylate adhesives ($n = 5$
fragments per treatment). Then, the fragment was indented by a Vickers 4-sided diamond pyramid indenter for 10 s in
the loading phase (Peak load: 300 mN; Creep: 2 s). In the unloading phase, the load decreased at the same rate as the
loading phase until the loading force became zero. At least five random locations on each fragment were indented.
Vickers hardness ($H$) and elastic modulus ($E$) were calculated based on the load-displacement curve using software
WIN-HCU (Fischer, Germany). Vickers hardness to elastic modulus ratio ($H/E$) was calculated to indicate the fracture
toughness of shells (Marshall et al., 1982). Organic matter content of the newly-produced shells collected from another
five individuals was determined by mass loss upon ignition at 550°C in a muffle furnace for six hours ($n = 5$ replicates
per treatment).

Given the limited amount of newly-produced shells, shells from three to five individuals from the same
treatment were powdered to make one composite shell powder sample as a replicate for the analyses of the following
geochemical properties. Shell powder was prepared by removing the newly-produced shells using a pair of forceps,
rinsing them with deionized water to remove the microalgae and other debris, drying them at room temperature and
finally grinding them into powder (particle size: ~5 μm) using a mortar and pestle. Carbonate polymorphs were
analysed using an X-ray diffractometer (D4 ENDEAVOR, Bruker, Germany). A small quantity of shell powder was
transferred onto a tailor-made sample holder and then scanned by Co Kα radiation (35 kV and 30 mA) from 20° to
70° 2θ with step size of 0.018° and step time of 1 s ($n = 3$ replicates per treatment). Carbonate polymorphs were
identified based on the X-ray diffraction spectrum using the EVA XRD analysis software (Bruker, Germany). Calcite
to aragonite ratio was calculated using the following equation (Kontoyannis and Vagenas, 2000):

$$\frac{I_C^{104}}{I_A^{221}} = 3.157 \times \frac{X_C}{X_A}$$

where $I_C^{104}$ and $I_A^{221}$ are the intensity of calcite 104 peak (34.4° 2θ) and aragonite 221 peak (54.0° 2θ), respectively;
$X_C/X_A$ is the calcite to aragonite ratio.

Magnesium to calcium ratio was determined by energy dispersive X-ray spectroscopy under the Philips XL30
field emission scanning electron microscope (Ries, 2004; Zhang et al., 2010; Leung et al., 2017b). A small quantity of
shell powder was transferred onto a stub and coated by carbon ($n = 3$ replicates per treatment; 3 trials per replicate).
The shell powder was irradiated by an electron beam with an accelerating voltage of 12 kV to obtain the energy
spectrum with background correction. Elements were identified and magnesium to calcium ratio was calculated using
software Genesis Spectrum SEM Quant ZAF (EDAX, USA). To determine relative amorphous calcium carbonate
(ACC) content, 1 mg shell powder was mixed with 10 mg potassium bromide, followed by compressing the mixture

into a disc (diameter: 13 mm) using a manual hydraulic press ($n$ = 3 replicates per treatment) (Chan et al., 2012). An infrared absorption spectrum ranging from 600 cm$^{-1}$ to 1800 cm$^{-1}$ with background calibration for the baseline was obtained using a Fourier transform infrared spectrometer (Avatar 370 DTGS, Nicolet, USA). The relative ACC content was estimated as the intensity ratio of the peak at 856 cm$^{-1}$ to that at 713 cm$^{-1}$ (Beniash et al., 1997).

**2.6 Statistical analysis**

Two-way permutational analysis of variance (PERMANOVA) was applied (number of permutations: 999; Euclidean distance calculated) to test the effects of hypoxia and non-lethal shell damage on the shell growth, fracture toughness, organic matter content, calcite to aragonite ratio, magnesium to calcium ratio, relative ACC content, respiration rate and clearance rate using software PRIMER 6 with PERMANOVA+ add-on (Anderson, 2001).

**3 Results**

*H. diramphus* had continuous shell growth throughout the 3-week exposure period, but the growth was faster after non-lethal shell damage (Fig. 2, Table A2). Hypoxia slightly, but significantly, hindered shell growth in both contexts. As a result, shell growth was the lowest for those undamaged individuals reared under hypoxia. The fracture toughness of newly-produced shells was enhanced by approximately two times after non-lethal shell damage (c.f. control), while hypoxia had no significant effect (Fig. 3, Table A2). As for the geochemical properties of newly-produced shells, organic matter content was elevated by ~2% after non-lethal shell damage, whereas the effect of hypoxia was indiscernible (Fig. 4a, Table A2). Calcite was the dominant carbonate polymorph and its proportion increased under hypoxia (Fig. 4b, Table A2). *H. diramphus* produced high-Mg calcite (i.e. Mg/Ca > 0.04) and the Mg/Ca in calcite increased to ~0.22 under hypoxia (Fig. 4c, Table A2). The relative ACC content was slightly elevated under hypoxia, meaning that less crystalline shells were produced (Fig. 4d, Table A2, Fig. A2 for the IR spectra). Calcite/Aragonite, Mg/Ca in calcite and relative ACC content were not significantly affected by non-lethal shell damage (Table A2). Regarding the physiological performance of *H. diramphus*, respiration rate was significantly reduced by both hypoxia and non-lethal shell damage (Fig. 5a, Table A2), but the impact of hypoxia was much greater. Clearance rate decreased significantly not only under hypoxia, but also after non-lethal shell damage under normoxia (Fig. 5b, Table A2). It is clear that the feeding rate of *H. diramphus* can be substantially impacted by either hypoxia or non-lethal shell damage.

**4 Discussion**

Hypoxia is expected to diminish the fitness and survival of marine organisms, probably leading to serious ramifications on marine ecosystems, such as changes in species populations, community structure and ecosystem functioning (Wu, 2002; Diaz and Rosenberg, 2008). Nevertheless, many less mobile marine organisms (e.g. molluscs,

polychaetes and echinoderms) are generally tolerant to hypoxia in the short term (Vaquer-Sunyer and Duarte, 2008), suggesting their potential capacity to accommodate its impacts. Despite the substantial reduction in respiration rate and feeding rate under hypoxia, we found that calcification and defence response of a calcifying polychaete were generally maintained, which could be associated with mineralogical plasticity, such as increased calcite to aragonite ratio and magnesium to calcium ratio.

Since energy demand for calcification is enormous mainly due to the production of organic matrix (Palmer, 1983, 1992), the reduction in energy gain by feeding and energy production by aerobic respiration under hypoxia would undermine both quality and quantity of shells produced by calcifying organisms (Cheung et al., 2008; Wijgerde et al., 2014). Under unthreatened conditions (i.e. without shell damage), we found that hypoxia slightly hinders the shell growth of *H. diramphus*. However, hypoxia did not affect the fracture toughness (i.e. mechanical strength) of newly-produced shells. The retarded shell growth under hypoxia could be pertinent to the reduced feeding rate, and hence energy reserves for calcification. While energy gain by feeding is suggested to be fundamental for shell growth (Melzner et al., 2011; Thomsen et al., 2013; Leung et al., 2017a), aerobic respiration is necessary to efficiently convert energy reserves into metabolic energy for various biological processes, including calcification. As such, the retarded shell growth is more likely ascribed to the hypoxia-induced metabolic depression, which reduces the amount of metabolic energy allocated to calcification. The quantity of organic matter (e.g. matrix proteins) occluded in the shell is a key factor affecting mechanical strength (Weiner and Addadi, 1997; Addadi et al., 2006; Marin et al., 2008). Since the organic matter content of newly-produced shells was not affected by hypoxia, mechanical strength can be maintained. Our results imply that similar amount of metabolic energy is allocated to the production of organic matter for shell strength, while less to inorganic components (i.e. calcium carbonate) for shell growth under hypoxia. This strategy (i.e. shell quality over shell quantity) is favourable under energy-limiting conditions because there is no exigency to expedite shell growth when risk is not imminent and the shell can already offer sufficient protection.

Under life-threatening conditions (i.e. following non-lethal shell damage), *H. diramphus* exhibited defence response, indicated by the production of tougher shells at a higher rate. As *H. diramphus* is sessile, enhancing the protective function of shells is probably the most effective defence response. Therefore, more organic matter was produced and occluded in the newly-produced shell to augment mechanical strength. Additionally, the carbonate crystals in the shell appeared to be more compacted (Fig. 6), which could also strengthen the shell. Such inducible defence response is commonly exhibited by calcifying organisms because shell repair should be prioritized to restore and enhance protection (Cheung et al., 2004; Hirsch et al., 2013; Brom et al., 2015). However, trade-offs are involved to activate defence response, such as reduction in the less essential biological processes or activities (Rundle and Brönmark, 2001; Trussell and Nicklin, 2002; Hoverman and Relyea, 2009; Babarro et al., 2016). For example, Brookes and Rochette (2007) showed that the calcification rate of a grazing gastropod is promoted under predation risk at the expense of grazing activity and somatic growth. Here, similar trade-offs were observed in *H. diramphus* (i.e. enhanced shell growth against reduced feeding rate). Indeed, when animals are under life-threatening conditions and the chance of survival becomes very low, they have to prioritize defence response (e.g. production of stronger shells for calcifying organisms) as the last resort to maximize survival rate (Bourdeau, 2009). This proposition is

corroborated by our results showing that *H. diramphus* allocated more metabolic energy not only to enhance shell growth, but also to synthesize more energy-demanding organic matrix to augment the mechanical strength of shells following non-lethal shell damage.

We expected that defence response would deteriorate under hypoxia in view of the substantial energy demand for shell production. Contrary to this prediction, *H. diramphus* can still produce tougher shells at a higher rate (c.f. Intact), meaning that the effect of hypoxia on defence response is mild in view of the slight impact on the shell growth. This unexpected finding not only reveals the strong tolerance of *H. diramphus* to hypoxia, but also suggests potential mechanisms that enable efficient calcification under hypoxia despite the reduced metabolic energy. We propose that

changing mineralogical properties could help compensate for the reduced metabolic energy in order to sustain defence response. In fact, the mineralogical properties of *H. diramphus* were altered consistently in response to hypoxia, irrespective of context. We found that hypoxia resulted in a greater proportion of calcite in the shell. When metabolic energy is reduced, precipitation of calcite is favourable because it requires less metabolic energy and allows faster shell growth than that of aragonite (Weiner and Addadi, 1997; Hautman, 2006; Ries, 2011). For instance, Ramajo et

al. (2015) showed that gastropod *Concholepas concholepas* increases calcite precipitation under metabolic depression; Chan et al. (2012) found that the calcite to aragonite ratio in the shell of polychaete *Hydroides elegans* is elevated at pH 7.4, which incurs metabolic cost for acid-base regulation. Apart from changing carbonate minerals, we found that more magnesium ions were incorporated into the newly-produced shell under hypoxia. It is evident that the incorporation of magnesium ions into calcite is actively regulated through various biological mechanisms, such as

active extrusion of excess magnesium ions at the calcification site (Bentov and Erez, 2006). The elevated Mg/Ca in calcite under hypoxia may suggest that the energy-requiring regulation of magnesium ions is reduced to conserve energy, which warrants further investigation. Furthermore, crystallization of amorphous calcium carbonate was slightly reduced by hypoxia, indicated by the higher relative ACC content. Since crystallization requires metabolic energy for the transport of carbonate ions (Addadi et al., 2006; Weiner and Addadi, 2011), our results suggest that

metabolic energy allocated to crystallographic control also decreased. Given the aforementioned changes in mineralogical properties, the energy cost for sustaining shell growth could be lessened. Such plastic response, also shown in some calcifying organisms under metabolic stress conditions (Ramajo et al., 2015; Leung et al., 2017a), may explain why the defence response of *H. diramphus* can generally be maintained under mild hypoxia in the short term. Interestingly, we found that such maintenance can last for at least three weeks, even though the energy intake by

feeding was markedly reduced by hypoxia. While the change in somatic tissue was not examined in this study, it is likely that *H. diramphus* consumes its energy reserves to enable the boosted shell growth (Palmer, 1983, Leung et al., 2013b).

        Despite the benefit of changing mineralogical properties as the plastic response, trade-offs against other phenotypic traits are inevitably incurred (Malausa et al., 2005; Leung et al., 2013b). For instance, shell solubility

increases due to the higher relative ACC content and Mg/Ca in calcite (Fernandez-Diaz, 1996; Ries, 2011; Fitzer et al., 2014). In other words, while the changes in mineralogical properties may allow sustained shell growth and

mechanical strength under hypoxia, the chemical stability of shells may be weakened. Nevertheless, our results suggest that the benefit of defence response probably outweighs the cost of this trade-off under life-threatening conditions.

Based on the present findings, we support the paradigm that calcification is mainly driven by the physiology of calcifying organisms rather than the seawater carbonate chemistry (Pörtner, 2008; Roleda et al., 2012). For example, the shell growth of *H. diramphus* decreased when the carbonate saturation state slightly increased under hypoxia. This is contradictory to the paradigm that calcification generally increases with carbonate saturation state, *vice versa* (Orr et al., 2005). Indeed, most calcifying organisms do not directly utilize carbonate ions, but bicarbonate ions, as the substrate for calcification, meaning that formation of calcareous shells is not a chemical reaction between calcium and carbonate ions (Pörtner, 2008; Roleda et al., 2012; Bach, 2015). This concept based on physiology explains why many calcifying organisms can maintain or even enhance calcification when carbonate saturation state is reduced (e.g. Ries et al., 2009; Garilli et al., 2015; Ramajo et al., 2016; Leung et al., 2017a).

Hypoxia can last for a long period of time (e.g. month) as observed in many coastal and marine open waters worldwide (Helly and Levin, 2004; Diaz and Rosenberg, 2008), and is predicted to be more prevalent in future due to ocean warming and human-induced eutrophication (Bijma et al., 2013). In order to maintain populations under hypoxia, calcifying organisms have to counter its impact on calcification. Despite the impaired aerobic metabolism, this study revealed that hypoxia only mildly hampers the shell growth of a calcifying polychaete, whereas its defence response (i.e. harder shells produced at a higher rate) can be sustained in the short term. This is likely mediated by modifying mineralogical properties of shells to reduce the energy demand for calcification. While some potential trade-offs are incurred, such plastic response could be the cornerstone of calcifying organisms to acclimate to metabolic stress conditions, and hence sustain their populations and ecological functions in coastal and marine ecosystems.

*Acknowledgements.* Financial support was provided by the University Grants Committee of Hong Kong Special Administrative Region (AoE/P-04/04) and the IPRS Scholarship from the University of Adelaide to JYSL. We acknowledge the staff in Adelaide Microscopy for their assistance.

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

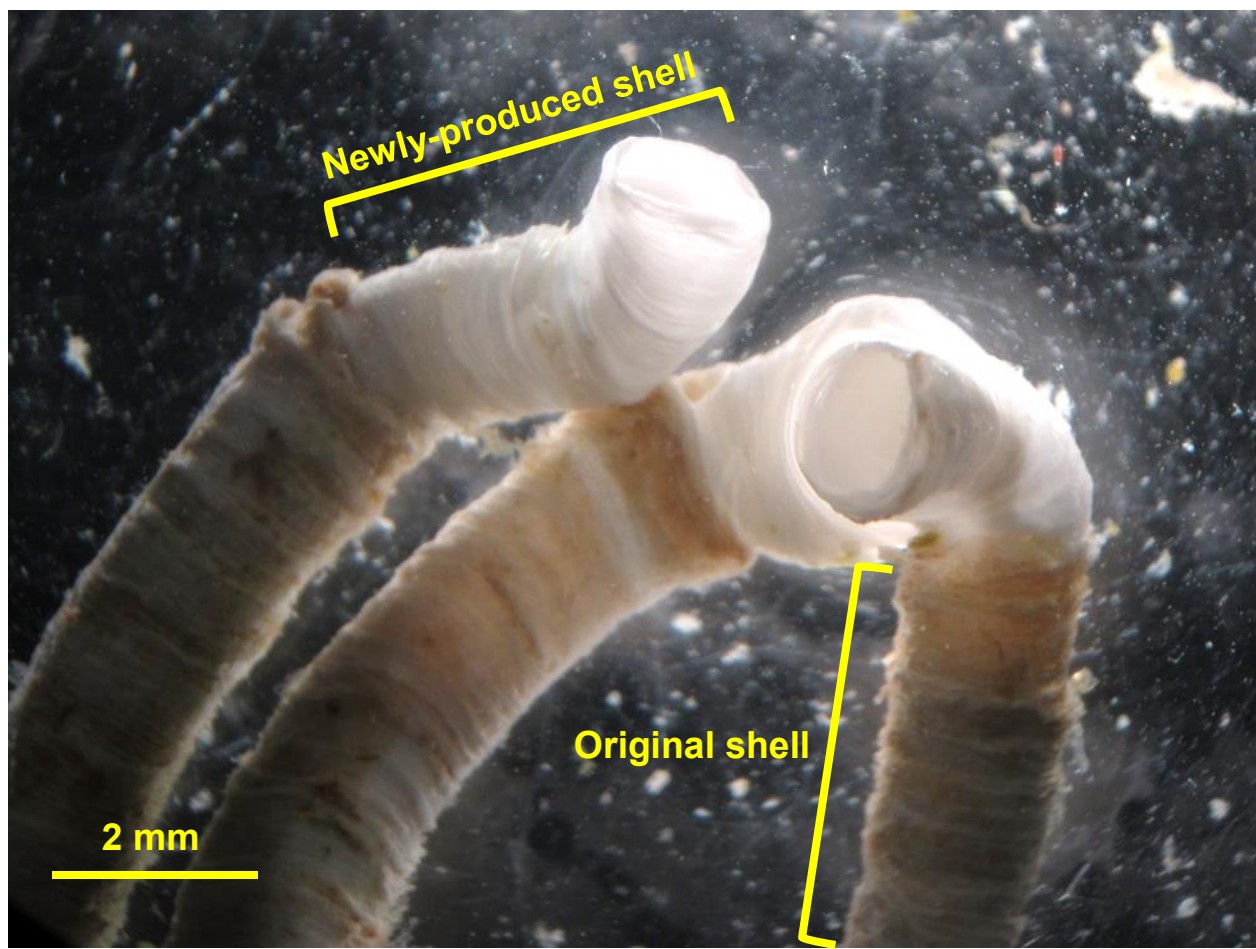

**Figure 1 A micrograph showing the newly-produced shell and original shell of *H. diramphus*, where the former is easily distinguished from the latter by the white colour. The original shell appears slightly coloured due to the biofilm (e.g. bacteria, algae, etc.) growing on the surface in the field.**

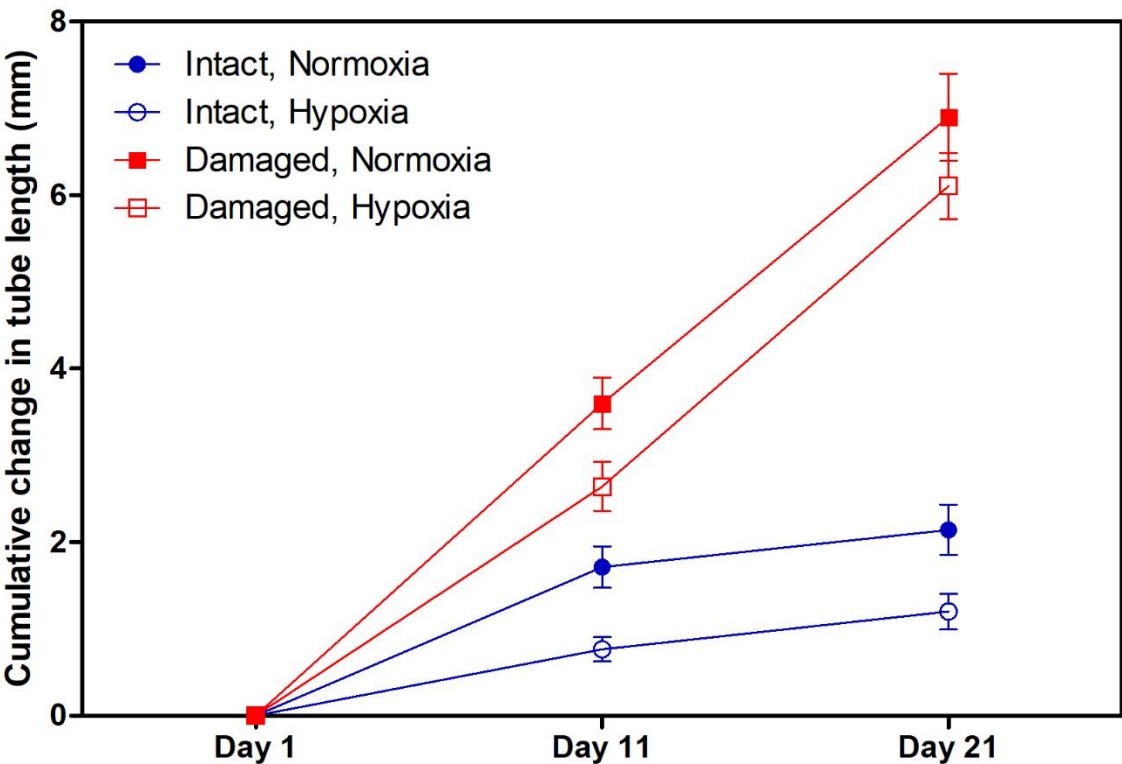

**Figure 2 Cumulative change in the tube length of *H. diramphus* in different treatments across the 3-week exposure period (mean ± S.E.; *n* = 30 for "Intact, Hypoxia" and "Damaged, Normoxia"; *n* = 28 for "Intact, Normoxia" and "Damaged, Hypoxia" due to the mortality).**

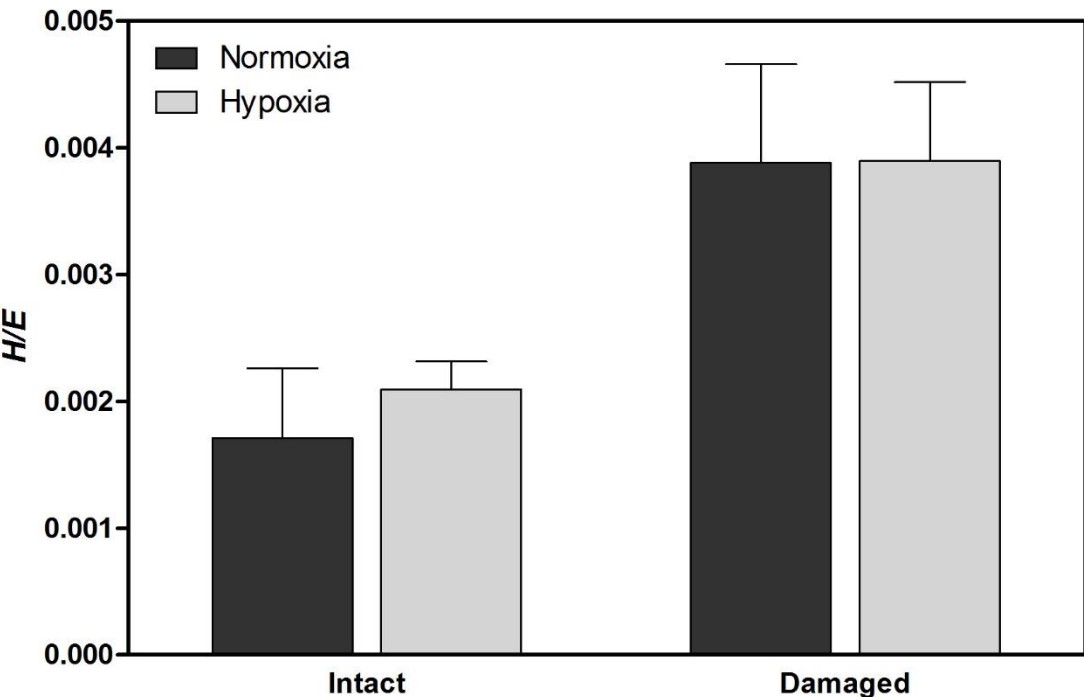

**Figure 3 Vickers hardness to elastic modulus ratio (*H/E*), indicating fracture toughness, of *H. diramphus* shells produced in different treatments (mean + S.E.; *n* = 5).**

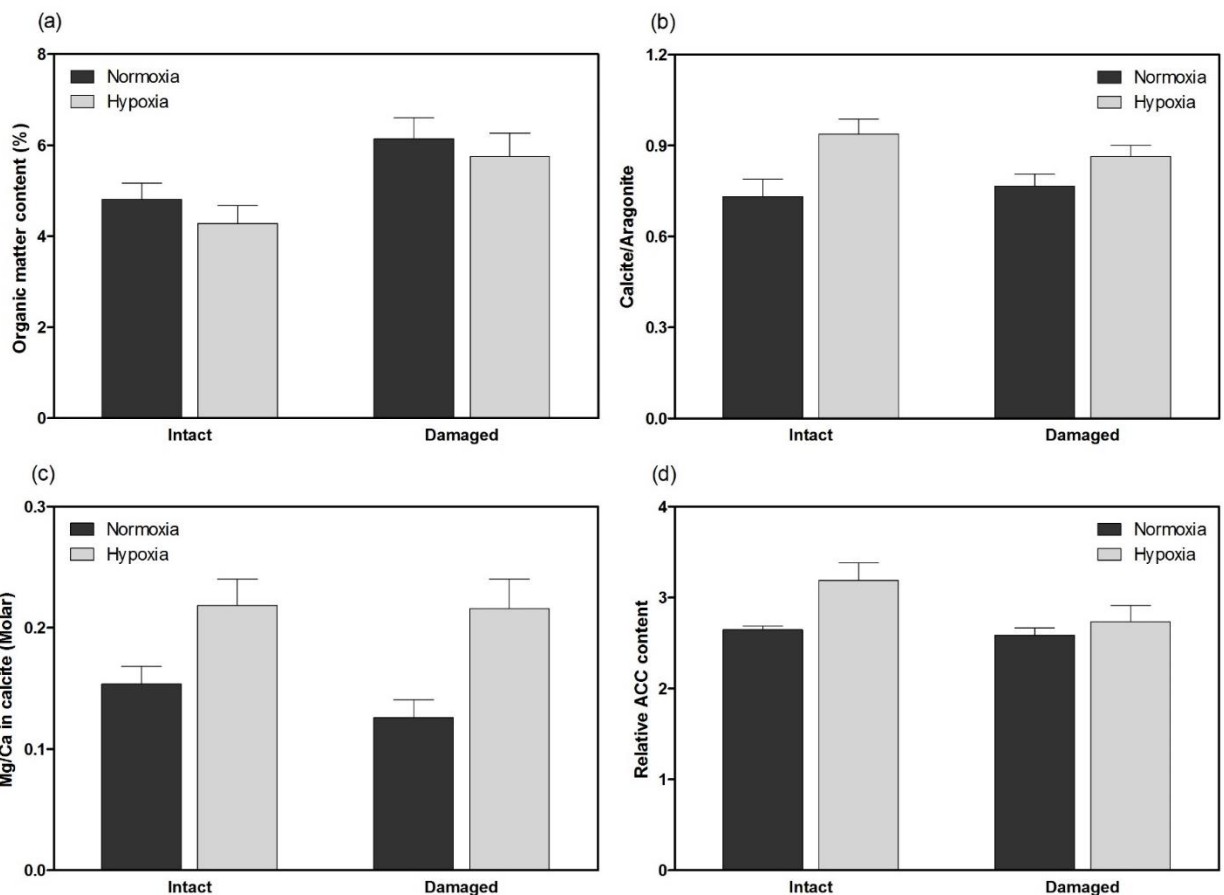

**Figure 4 Geochemical properties of *H. diramphus* shells, including (a) organic matter content, (b) calcite to aragonite ratio, (c) magnesium to calcium ratio in calcite and (d) relative amorphous calcium carbonate content, in different treatments (mean + S.E.; *n* = 3, except *n* = 5 for organic matter content).**

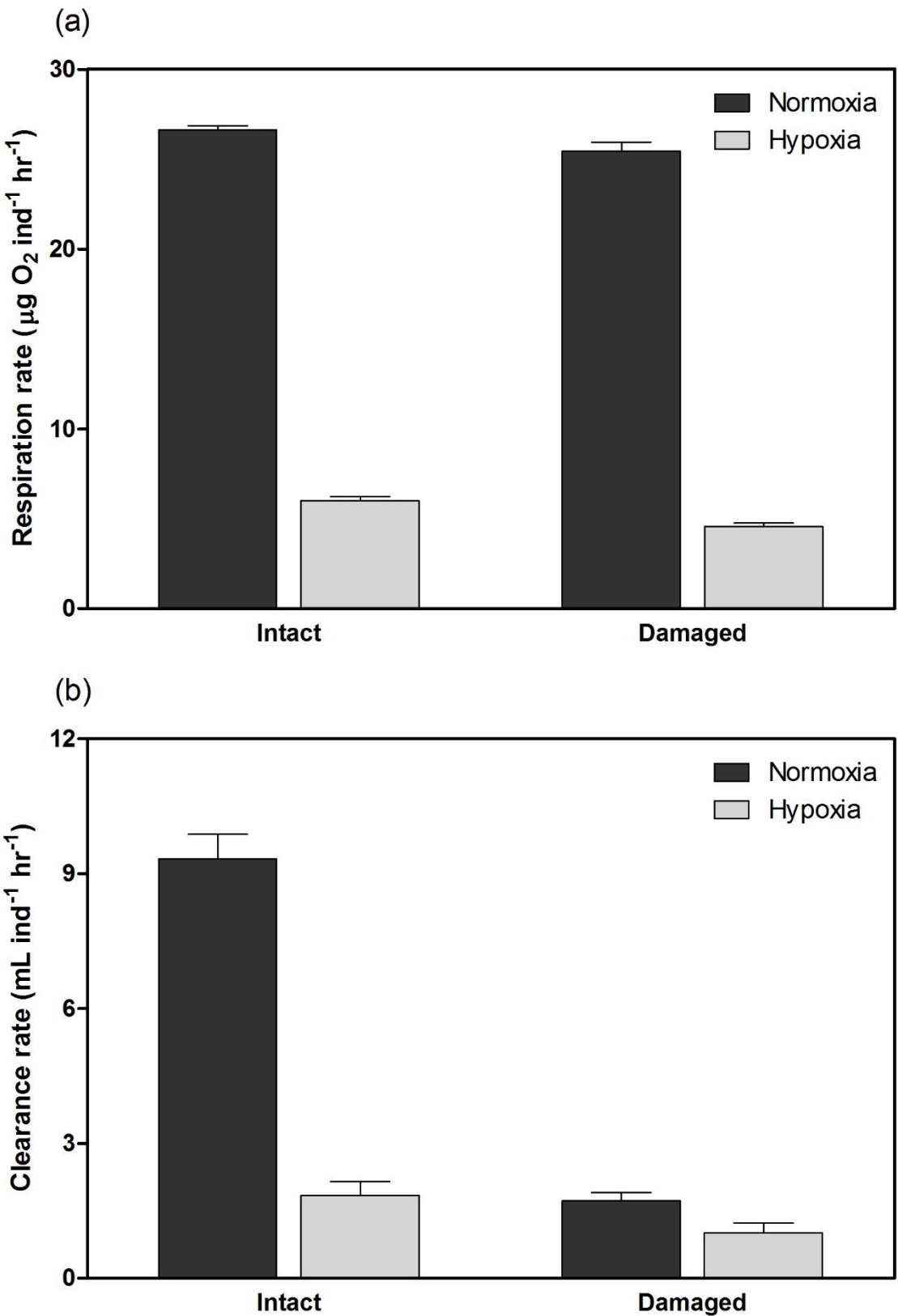

**Figure 5 (a) Respiration rate and (b) clearance rate of *H. diramphus* in different treatments (mean + S.E.; *n* = 5).**

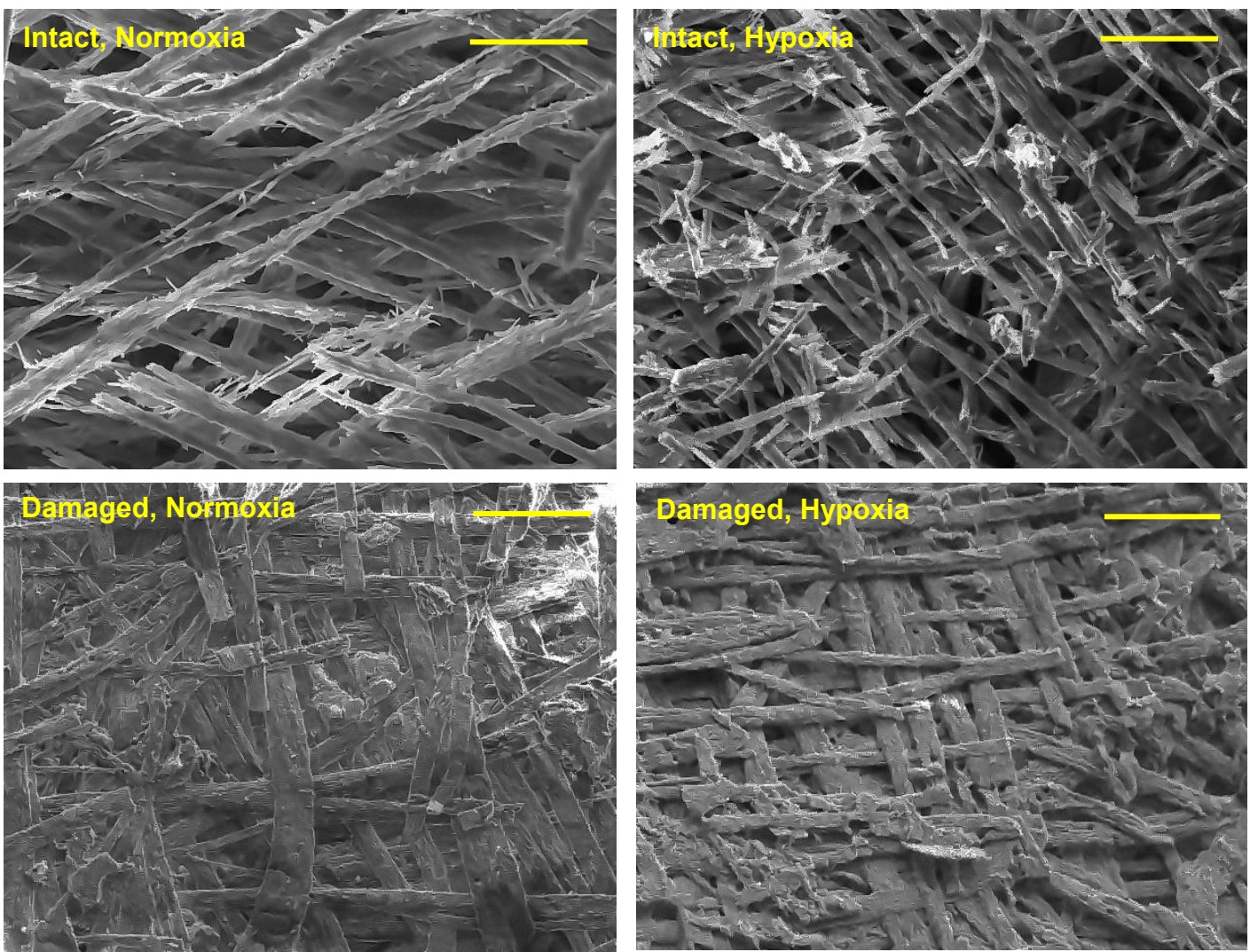

**Figure 6 SEM images of the inner surface of *H. diramphus* shells produced in different treatments, indicating the shell integrity. The carbonate crystals of newly-produced shells appear to be thicker and more compact following non-lethal shell damage, regardless of the dissolved oxygen level. Scale bar: 20 µm.**

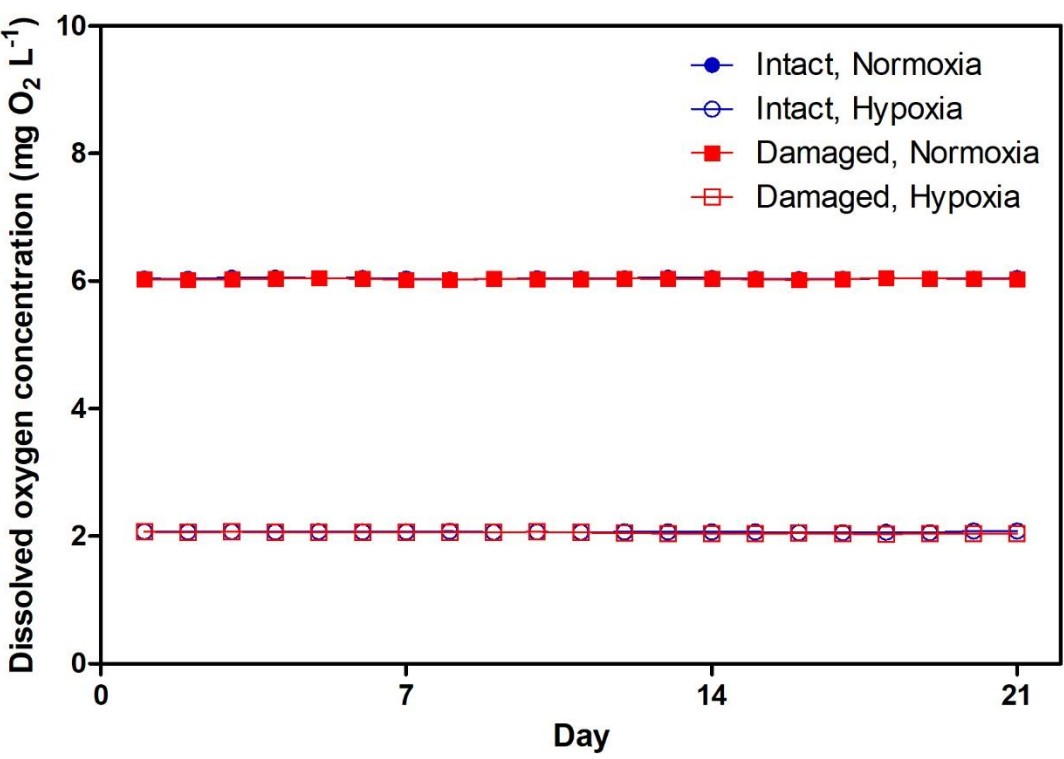

**Figure A1 Dissolved oxygen concentration of seawater in different treatments across the 3-week experimental period (mean ± S.D., $n = 3$).**

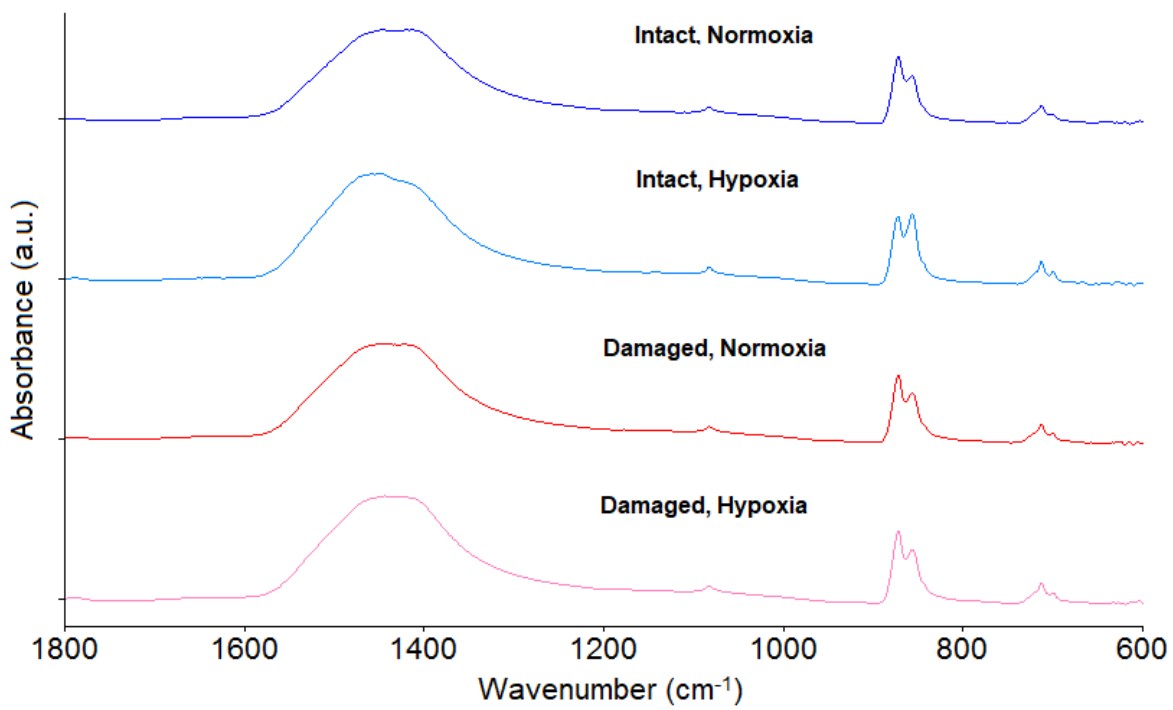

**Figure A2 Infrared spectra for the newly-produced shells of *H. diramphus* growing under different treatment conditions.**

**Table A1 The seawater parameters under different treatment conditions throughout the exposure period (mean ± S.D.). Dissolved oxygen concentration was daily measured using an optical dissolved oxygen probe (SOO-100, TauTheta Instruments, USA). pH was daily measured using a pH meter (HI 9025, HANNA Instruments, USA). Temperature and salinity were measured daily using a thermometer and refractometer, respectively. Total alkalinity was measured weekly using a titrator (HI 84431, HANNA Instruments, Germany). Saturation states (Ω) of calcite and aragonite were calculated using the CO2SYS program (Pierrot et al., 2006), with dissociation constants from Mehrbach et al. (1973) refitted by Dickson and Millero (1987).**

| | Intact, Normoxia | Intact, Hypoxia | Damaged, Normoxia | Damaged, Hypoxia |
|---|---|---|---|---|
| Measured parameters | | | | |
| Dissolved oxygen (mg $O_2$ $L^{-1}$) | 6.04 ± 0.02 | 2.07 ± 0.03 | 6.03 ± 0.01 | 2.05 ± 0.03 |
| pH (NBS scale) | 8.10 ± 0.05 | 8.26 ± 0.04 | 8.09 ± 0.05 | 8.26 ± 0.04 |
| Temperature (°C) | 28.2 ± 0.08 | 28.2 ± 0.09 | 28.2 ± 0.08 | 28.2 ± 0.10 |
| Salinity (psu) | 32.9 ± 0.35 | 33.0 ± 0.25 | 33.0 ± 0.42 | 33.1 ± 0.25 |
| Total alkalinity ($\mu$mol $kg^{-1}$) | 2241 ± 8.96 | 2231 ± 12.2 | 2241 ± 9.11 | 2243 ± 9.44 |
| Calculated parameters | | | | |
| $C_T$ ($\mu$mol $kg^{-1}$) | 1984 ± 26.1 | 1885 ± 27.9 | 1988 ± 29.0 | 1895 ± 24.3 |
| $HCO_3^-$ ($\mu$mol $kg^{-1}$) | 1784 ± 40.2 | 1632 ± 44.3 | 1790 ± 45.0 | 1641 ± 38.5 |
| $CO_3^{2-}$ ($\mu$mol $kg^{-1}$) | 187 ± 16.1 | 244 ± 17.6 | 184 ± 18.3 | 246 ± 15.4 |
| $\Omega_{calcite}$ | 4.61 ± 0.40 | 6.01 ± 0.43 | 4.54 ± 0.45 | 6.05 ± 0.38 |
| $\Omega_{aragonite}$ | 3.04 ± 0.26 | 3.98 ± 0.29 | 3.01 ± 0.30 | 4.01 ± 0.25 |

Operation manual of titrator for total alkalinity:

https://www.manualslib.com/manual/530078/Hanna-Instruments-Hi-84431.html#manual

Operation manual of optical dissolved oxygen probe for dissolved oxygen concentration:

https://in-situ.com/wp-content/uploads/2015/05/Stable_Optical_Oxygen_System_-SOO-100_Manual.pdf

**Table A2 PERMANOVA table showing the effects of dissolved oxygen (DO) and context on shell growth, fracture toughness, organic matter content, calcite/aragonite, Mg/Ca in calcite, relative ACC content, respiration rate and clearance rate.**

| | $df$ | Mean square | Pseudo-F | $p$ | Comparison of means |
|---|---|---|---|---|---|
| Shell growth (Day 21) | | | | | |
| DO | 1 | 21.8 | 5.66 | **0.019** | Normoxia > Hypoxia |
| Context | 1 | 676 | 175 | **0.001** | Damaged > Intact |
| DO × Context | 1 | 0.163 | 0.042 | 0.838 | |
| Fracture toughness | | | | | |
| DO | 1 | $2.00 \times 10^{-7}$ | 0.119 | 0.734 | |
| Context | 1 | $1.97 \times 10^{-5}$ | 11.7 | **0.004** | Damaged > Intact |
| DO × Context | 1 | $1.69 \times 10^{-7}$ | 0.101 | 0.755 | |
| Organic matter content | | | | | |
| DO | 1 | 1.04 | 1.10 | 0.309 | |
| Context | 1 | 9.83 | 10.5 | **0.005** | Damaged > Intact |
| DO × Context | 1 | 0.022 | 0.024 | 0.880 | |
| Calcite/Aragonite | | | | | |
| DO | 1 | 0.070 | 11.0 | **0.017** | Hypoxia > Normoxia |
| Context | 1 | $1.14 \times 10^{-3}$ | 0.178 | 0.623 | |
| DO × Context | 1 | $8.84 \times 10^{-3}$ | 1.39 | 0.249 | |
| Mg/Ca in calcite | | | | | |
| DO | 1 | 0.018 | 16.0 | **0.004** | Hypoxia > Normoxia |
| Context | 1 | $6.92 \times 10^{-4}$ | 0.618 | 0.455 | |
| DO × Context | 1 | $4.83 \times 10^{-4}$ | 0.431 | 0.530 | |
| Relative ACC content | | | | | |
| DO | 1 | 0.355 | 6.02 | **0.047** | Hypoxia > Normoxia |
| Context | 1 | 0.199 | 3.37 | 0.104 | |
| DO × Context | 1 | 0.116 | 1.97 | 0.206 | |
| Respiration rate | | | | | |
| DO | 1 | $2.15 \times 10^{-3}$ | $4.36 \times 10^{3}$ | **0.001** | Normoxia > Hypoxia |
| Context | 1 | $8.52 \times 10^{-6}$ | 14.2 | **0.001** | Intact > Damaged |
| DO × Context | 1 | $7.40 \times 10^{-8}$ | 0.150 | 0.715 | |
| Clearance rate | | | | | |
| DO | 1 | 84.0 | 140 | **0.001** | Within Intact: Normoxia > Hypoxia<br>Within Damaged: Normoxia > Hypoxia |
| Context | 1 | 89.1 | 148 | **0.001** | Within Normoxia: Intact > Damaged<br>Within Hypoxia: N.S. |
| DO × Context | 1 | 57.2 | 95.5 | **0.001** | |