# Peer review of "Effects of hypoxia and non-lethal shell damage on shell mechanical and geochemical properties of a calcifying polychaete"

_Biogeosciences, 2017_

## Referee Comment (RC1) · Anonymous Referee #1 · 12 Oct 2017

Dear Editor,

The manuscript "Calcification and inducible defense response of a calcifying organism could be maintained under hypoxia through phenotypic plasticity" by Leung and Cheung, presents interesting questions about possible eco-physiological adaptations observed on a calcifying polychaete exposed to acute hypoxia, including changes in calcification rates, shell composition and metabolism.

Despite some interesting points, I think that the manuscript, in the present form is not acceptable for publication, because of a substantial lack of detail on the protocols used for the experiments and the analyses, very shallow description of the main results and some over-interpretation of the results. A (very) major revision is therefore suggested.

Here listed some of the **major comments**:

1) The materials and methods are too undetailed. Are the specimens at the T0 adult or juveniles? Were they exposed to a day/night light cycle? If so, how did you control algal proliferation during the experiment? If not, are you sure that this does not influence their physiology? Were added algae dead or alive? What basis the 20 ml algal concentration was chosen on? Before organic matter analyses, were the samples washed? If so, how? Also for statistical analyses more details are needed.

2) I do not understand why in the materials and methods the authors say that they used 10 specimens per replicate (x 3 repl.) per treatment, but each observed response in the results is based on only n=3 or n=5 specimens (or replicates??). This is surprising if we consider that most of the performed analyses (e.g. respiration rates, growth rates, clearance rate) are not destructive and that just part of the shell is necessary for the rest of the analyses (e.g. Mg/Ca, organic matter, ACC...). So why did not the author use more than 3 or 5 specimens to do their analyses and calculate averages? Could they add more replicates?

3) The experimental design is quite weak. Maybe this impression derives also from the difficulty to understand how many individuals/measures/replicates where performed for each parameter. We lack fundamental information about the ability of these organisms to develop into cultures. All the results are discussed on the basis of comparison to normoxic conditions. How can the authors be sure that a "culturing effect" is not interfering with the results?
This is particularly true for shell composition. Why did not the author compare the shells of individuals grown in normoxic conditions to pre-experiment portions of shells (grown into natural conditions) to see if any "culturing effect" is visible?

4) The protocol used for respiration measurements is unusual to me. I cannot understand how the authors did measure the oxygen content of hypoxic waters into a syringe with a relatively thick probe tip and can be sure they avoided oxygenation during the measurement. Also, are they sure that the material the syringe is made of is impermeable to oxygen? Do they have any measure of blanks to estimate the possible gas exchange through the syringe walls during the analysis?

5) In the experiment the authors consider hypoxic conditions to be reached at ~2.0 mg $O_2$/L. This concentration is the one which is normally considered to be the upper limit for hypoxia. Considering that the error associated to the probe used for $O_2$ survey is 0.1 mg/L (source: TauTheta manual) and that the average values the authors report in table S1 are always a slightly above 2.0 mg/L for hypoxic conditions, I wonder why the authors did not test a lower

concentration to be sure to never exceed oxygen concentrations corresponding to namely hypoxic conditions. The results obtained should be carefully presented as the response to a case of acute (short time) and slight (upper limit) hypoxia and every over-interpretation or generalization to strong or long-term hypoxia should be avoided.

6) In lines 96-97 authors say that part of the individuals was left into hypoxic conditions during one week before the experiment (after damage and measures of the tubes). Can you please justify this choice? Does this mean that the T0 for tube sizes represents instead one week under treatment conditions? Did the authors measure the T0 size again after the acclimation and before the start of the experiment?

7) How did the authors get to have stable oxygen conditions during and just after water changes (every 3 days)? Did you measure oxygen into the culture solution before retiring old seawater? Could you please show these data somewhere in supplementary materials?

8) Is edx analysis resolution enough for consistent Mg/Ca measurement in the shell? Can the authors give more information about the accuracy, precision, detection limits etc. of the analysis, please? Or add references if the protocol is routinely used.

9) Some of the conclusions/discussions are inferential and not supported by the data. For example paragraph from line 249 to 258 should be deleted, in my opinion, as it is not supported by the presented data.

10) The hypothesis of relaxed magnesium regulation to explain higher Mg/Ca in the calcite produced under hypoxia is based on benthic foraminifera. These organisms are known to strongly discriminate against magnesium. This does not seem the case for polychaetes, which seem to contain very high concentrations of Mg in the shell. Authors should base their hypotheses on more adapted literature.

**Minor comments/suggestions:**

Line 30: use "increase" instead of "augment"

Line 51: replace "It" by "This"

Line 53: What does "or other physiological processes via energy trade-off" mean? Can you explain what other processes you're talking about please?

Line 58: Please delete "and defense response".

Materials and methods: Please add titration protocol for alkalinity measures presented in table S1.

Lines 93-97: Please add more details about the procedure used to measure the specimens, associated errors and discuss the potential stress that this manipulation may represent and the effect it could have on the final results

The order of paragraphs in the Material and methods section is, at present, a bit confusing and should follow a more logical pattern. I would suggest that the paragraph on experimental design should show the setting, the replication, times etc., for all type of analyses. Then a paragraph on procedures for physiological analyses should explain all the used methods for respiration rates, survival rates, feeding rates and shell growth. Then a final paragraph on the shell composition should follow.

Lines 99-100: Why is the color of new shell very different from the ancient one? Is it normal? Maybe you should discuss it somewhere. One would think that it is a culturing effect on shell structure... Figure S1 should be in the main text.

Lines 103-112 should be part of the "Experimental set up" paragraph

Line 123: The specimens used for organic matter composition are the same measured for toughness? Please detail this kind of information

Paragraph 123-129: 1) for the composite shell power did you mix calcite from different specimens, isn't it? Did they come from a same replicate for a treatment or even replicates were mixed? How did you prepare the powder exactly? Did you wash the shell before to avoid contamination (from algae given as food for example)? How? Why did you choose Mg/Ca ratio as a parameter to be measured?

Paragraph 148-155: How did you calibrate oxygen probes before the analysis?

Line 152: "The air inside the syringe...": what air? Why is it there during the first measure? This part is not clear to me.

Line 154: "by gently stirring the FSW inside the syringe". How did you avoid oxygenation at this step?

Line 156: Please specify the unit used for consumed oxygen. Normally mL or μmol are used. In your figure 4a you use μg $O_2$, which is quite unusual as a unit for oxygen.

Lines 159-160: Why did you only used one algal species for this experiment? Can you add fundamental details such as if the experiment was performed under light conditions, please? Also you say that 5 replicated bottles were used per treatment. In the first materials and methods section you say you have 3 replicated bottles with 10 specimens for the experiments. I'm lost... Are these different bottles? How many individuals per bottle per treatment do you have then?

Statistical paragraph: Where data transformed before the analyses to homogenize the magnitudes? How many permutations were performed? Which distance parameter was used and why? Can you specify the "aforementioned parameters" of line 168, please?

Line 172: You say hypoxia slightly hindered, but your statistics say this difference is significant, so maybe this should be emphasized a bit more.

Line 174: Please replace "negligible" with "no".

Line 180: You say "but only slightly by non-lethal shell damage". It looks like a significant difference, visually (fig. 4a). Is it confirmed by statistical analyses? Yes, so you should say it!

Lines 181-182: specify whether statistical differences are visible and where.

Line 187: what do you mean by "ramifications"?

Line 188: "tolerant to hypoxia", at what temporal scale?

Line 194: I would suggest not to use "unthreatened conditions" as a general for "shell damage", because hypoxia also is an unthreatened condition, so it can result confusing.

Line 195: "hypoxia slightly hinders"... again, is it significant or not?

Lines 201-202: redundant concept, already said.

Lines 205-206: Please replace "shell growth" with "inorganic components of the shell". You suggest that under hypoxia the production of organic matter compensate for diminished quality of inorganic components (>ACC). Palmer (1992) on the contrary suggests that organic matter production is costly and that would be the reason why high-organic calcareous microstructures became rare with evolution. How do you explain this incoherence?

Line 211-212: Could the overproduction of organic matter be related to higher calcification rates?

Lines 219-221: I do not understand what you mean

Line 224: You say the effect of hypoxia on defense response is not discernible. Although toughness is not affected, I would say that reduced shell growth rates (visible in figure 1) should be taken into account, to discern the effects of hypoxia, as well.

Line 234: please replace "signifies" with "may suggest".

Line 241: At the end of the sentence you should add ".. can generally be maintained at least under slight hypoxia, on a short timescale."

Line 248: please delete "and therefore *H. diramphus* prioritized defense response".

Line 259: please add "open" between marine and waters (because "costal" are also marine waters!)

Line 264: when you say that the defense response can be sustained you should specify "on a short timescale", because your results are based on short-term experiments.

Lines 330-339: Please order the references on a chronological basis.

Line 338 and 368: add authors to the list

Survival rate figure (S2) should not be in supplementary material, as this is an important result to take into account in the analysis of all the others.

---

## Referee Comment (RC2) · Anonymous Referee #2 · 18 Oct 2017

The manuscript by Leung and Cheung provides information regarding how calcification processes in polycheate worms could be influenced by future hypoxia. The results are pretty straight forward, and I consider these types of studies are important, although not ground-breaking. I have a few concerns that should be addressed before this manuscript could be accepted.

The grammar and style should be improved in the introduction and discussion of this m/s before it could be published in any outlet. There are too many examples of this for me to highlight every one, but for example the use of the term "defence response".

[Figure]

I suggest the authors ask a senior colleague who is a native English speaker to read over and correct for them. In general there is also a lot of speculation for a 21 day long study.

Specific comments: Line 40: In general I agree that calcification costs energy, but in some organisms the energy-dependence has been postulated as low (e.g. in corals – see McCulloch et al. (2012)). So this may be true for gastropods, but not necessarily so for some other organisms. So this sentence needs to be balanced somewhat. Line 70: The hypotheses around phenotypic plasticity needs to be strengthened and clarified. What exactly is the phenotype that is plastic here? The capacity to form different types of mineral in the shell? Or simply that responses will differ between control and reduced O2 concentrations? Reading the discussion, I think the authors are misusing the term phenotypic plasticity. Demonstrating variability in responses of individuals within the same population to a stressor is not demonstrating phenotypic plasticity, nor is demonstrating a different response under different treatments between different individuals. Line 82: How was pH measured, and on what scale, using what buffers? More information needed here. How was salinity and temperature measured? I see some of these details in table S1, but there are required in the methods section. Statistical analysis: why use a permanova? I would expect each parameter to be separated analysed using univariate analyses as a first step. A justification for using permanova over an anova or linear model needs to be justified here. Line 191-192: But this statement is at odds with the findings of the permanova and the figures, that calcification was impacted by hypoxia in this study. Also, the end of the sentence that this could be due to phenotypic plasticity needs to be explained, as this makes no sense to me.

References used in the review: McCulloch MT, Falter J, Trotter J, Montagna P (2012) Coral resilience to ocean acidification and global warming through pH up-regulation. Nature Climate Change, 2, 623-627.

---

## Referee Comment (RC3) · Anonymous Referee #3 · 20 Oct 2017

Major Comments:

Title: I do not find this title representative of the authors results/discussion. Please describe which aspects of the phenotype are considered plastic, since there is no change in mechanical strength and authors discuss less crystallographic control and magnesium regulation under hypoxic conditions.

Lines 34-38: Authors seem to go around a point here. Please state exactly what drives calcification instead of pH and seawater chemistry.

Line 54: Energy costs can increase rather than reduce, if organic content is modified such that it increases.

Line 76: Mean size of polychaetes?

Line 102: Diameter of hole drilled?

Line 103: Tubes were glued using what?

Line 114: How was the shell fragment acquired and cleaned of organic tissue?

Line 116: Was the same surface (e.g: inner shell surface) always used for indentation?

Line 122: Organic content of shell or whole polychaete? It is of the shell I assume, but is unclear.

Line 126: How was the shell powder acquired?

Line 130: Please provide information on how these polymorphs are typically distributed in the organism. Are the polymorphs specific to the outer/inner layer of the shell?

Line 139: Were ACC, aragonite and calcite standards measured? Please explain why, if not.

Line 140: Diameter of the KBR-shell powder disc?

FTIR: FTIR is a bulk measurement and ideally should not be used to infer "relative" proportions of carbonate polymorphs. Typically, the presence of a 713 cm$^{-1}$ peak is indicative of crystalline calcium carbonate comprising the bulk of shell carbonates. However, I am aware that this interpretation has been used before and if authors proceed with the analyses, could they please clarify if the spectra were scaled so that 713 cm$^{-1}$ peaks had the same heights as described in Weiss et al (2002)? In addition, please specify the typical size of crystallites in shell since such ratios have been demonstrated to be influenced by particle size (Kristova et al 2015).

Line 148: What were the syringes made to of?

Line 156: Hunger is only standardised if individuals were at the same start point.

Lines156-166: This doesn't represent clearance rates during the experiment.

Line 170-180: Please provide full FTIR spectra as a supplementary figure.

Lines 267-268: Can inferences be made regarding whether inner/outer layers were calcified if the polymorphs are specific to a layer of the polychaete shell?

Line 233-234: This is a strong statement. Regulation of Mg may be interpreted but the authors results do not **signify** that it is relaxed under hypoxia.

Line 256-257: Please delete this final sentence. It is a very strong statement and the whole paragraph does not explain why hypoxia the key stressor in the future (which is debatable anyway).

Minor Comments:

Line 10-11: Sentence like this needs a reference.
Line 25: change "shells" to skeletons.
Line 32: Delete "however".
Line 368: Please provide full reference.

Figure 5 (SEM): Are these images of the aragonite or calcitic parts of the shell? The legend needs more descriptive text. It is not obvious to me how these images indicate shell integrity.

Table A1: Please include other calculated parameters such as $HCO_3^-$, $CO_3^{2-}$ and $C_T$.

References used for review:

Weiss et al (2002) Mollusc larval shell formation: amorphous calcium carbonate is a precursor phase for aragonite. DOI: 10.1002/jez.90004

Kristova et al (2015) The effect of the particle size on the fundamental vibrations of the [CO3(2-)] anion in calcite. DOI: 10.1021/acs.jpca.5b02942.

---

## Author Comment (AC2) · 15 Dec 2017

Dear Editor,

The manuscript "Calcification and inducible defense response of a calcifying organism could be maintained under hypoxia through phenotypic plasticity" by Leung and Cheung, presents interesting questions about possible eco-physiological adaptations observed on a calcifying polychaete exposed to acute hypoxia, including changes in calcification rates, shell composition and metabolism.

Despite some interesting points, I think that the manuscript, in the present form is not acceptable for publication, because of a substantial lack of detail on the protocols used for the experiments and the analyses, very shallow description of the main results and some over-interpretation of the results. A (very) major revision is therefore suggested.

RESPONSE: We thank this reviewer for reviewing and providing useful suggestions for improving our manuscript. After reading the comments carefully, we found that most of them are related to the methods and can be clarified easily. However, there are some invalid arguments which are explained below. Overall, we are happy to make a major revision and believe that incorporating the good suggestions can greatly improve the quality of our manuscript.

Here listed some of the **major comments**:

1) The materials and methods are too undetailed. Are the specimens at the T0 adult or juveniles? Were they exposed to a day/night light cycle? If so, how did you control algal proliferation during the experiment? If not, are you sure that this does not influence their physiology? Were added algae dead or alive? What basis the 20 ml algal concentration was chosen on? Before organic matter analyses, were the samples washed? If so, how? Also for statistical analyses more details are needed.

RESPONSE: The specimens were adults and exposed to a day/light cycle of 14:10 hrs. Live algae were added, but there was no need to control their growth because the consumption rate of polychaetes is faster than the proliferation rate of algae. That is why we had to add algal suspension on a daily basis to sustain the growth of polychaetes (Ln 81-83). The feeding regime was based on our extensive experience in rearing *Hydroides* spp., which can ensure their normal growth under laboratory conditions. The shells were rinsed with deionized water before organic matter analysis. The above information will be added in the revision and more details about statistical analysis can be provided as suggested below.

2) I do not understand why in the materials and methods the authors say that they used 10 specimens per replicate (x 3 repl.) per treatment, but each observed response in the results is based on only n=3 or n=5 specimens (or replicates??). This is surprising if we consider that most of the performed analyses (e.g. respiration rates, growth rates, clearance rate) are not destructive and that just part of the shell is necessary for the rest of the analyses (e.g. Mg/Ca, organic matter, ACC...). So why did not the author use more than 3 or 5 specimens to do their analyses and calculate averages? Could they add more replicates?

RESPONSE: We adhere to the classic, unambiguous definition of "n" as "number of replicates" (i.e. not "number of individuals"). To clarify the sample size used, we initially had a total of 30 individuals per treatment (Ln 101-102, 105). Considering the mortality following the 3-week exposure, 25 individuals were made into 5 replicates with 5 individuals per replicate for respiratory rate and clearance rate measurements (Ln 149 and 158). Please note that one individual per replicate is unable to give adequate precision and reliability of measurements. Therefore, multiple individuals were pooled and each replicate is represented by an average of the pooled individuals (the units of these variables were expressed as "per individual"). Based on our experience and preliminary test, this design can already give adequate precision of measurement and statistical power. This is also evident from the small standard errors in our results (see Fig. 1 and 4).

3) The experimental design is quite weak. Maybe this impression derives also from the difficulty to understand how many individuals/measures/replicates where performed for each parameter. We lack fundamental information about the ability of these organisms to develop into cultures. All the results are discussed on the basis of comparison to normoxic conditions. How can the authors be sure that a "culturing effect" is not interfering with the results? This is particularly true for shell composition. Why did not the author compare the shells of individuals grown in normoxic conditions to pre-experiment portions of shells (grown into natural conditions) to see if any "culturing effect" is visible?

RESPONSE: Please see the above response for the issue of sampling design. As stated in the Introduction, this study aims to elucidate the effects of hypoxia on calcification and defence response of a calcifying polychaete, which can be experimentally tested under fully controlled laboratory conditions in order to isolate the effects of hypoxia and minimize the number of confounding factors. In contrast, the "culturing effect" cannot answer our research question because it only indicates the effects caused by the difference between laboratory and field conditions, but not the effects of hypoxia. As all the individuals were reared under laboratory conditions, the results were not influenced (or biased) by "culturing effect".

4) The protocol used for respiration measurements is unusual to me. I cannot understand how the authors did measure the oxygen content of hypoxic waters into a syringe with a relatively thick probe tip and can be sure they avoided oxygenation during the measurement. Also, are they sure that the material the syringe is made of is impermeable to oxygen? Do they have any measure of blanks to estimate the possible gas exchange through the syringe walls during the analysis?

RESPONSE: We have to invalidate this comment. Our DO probe is thin (~2.4 mm in diameter) enough to perfectly insert into the hole of plastic syringe (~2.7 mm in diameter) for DO measurement. The syringes used were ordinary plastic syringes (Terumo Hypodermic Syringe without a needle) with a barrel wall which is definitely thick enough to be impermeable to air. Once the syringe is sealed (Ln 151-152), oxygenation cannot occur under such air-tight conditions. This simple method has been widely used before (e.g. Zhao et al., 2011; Leung et al., 2013).

Leung, Y.S., Shin, P.K.S., Qiu, J.W., Ang, P.O., Chiu, J.M.Y., Thiyagarajan, V., and Cheung, S.G.: Physiological and behavioural responses of different life stages of a serpulid polychaete to hypoxia. Mar. Ecol. Prog. Ser., 477, 135–145, 2013.

Zhao, Q., Cheung, S.G., Shin, P.K.S., and Chiu, J.M.Y.: Effects of starvation on the physiology and foraging behaviour of two subtidal nassariid scavengers. J. Exp. Mar. Biol. Ecol., 409, 53–61, 2011.

5) In the experiment the authors consider hypoxic conditions to be reached at ~2.0 mg $O_2$/L. This concentration is the one which is normally considered to be the upper limit for hypoxia. Considering that the error associated to the probe used for $O_2$ survey is 0.1 mg/L (source: TauTheta manual) and that the average values the authors report in table S1 are always a slightly above 2.0 mg/L for hypoxic conditions, I wonder why the authors did not test a lower concentration to be sure to never exceed oxygen concentrations corresponding to namely hypoxic conditions. The results obtained should be carefully presented as the response to a case of acute (short time) and slight (upper limit) hypoxia and every over-interpretation or generalization to strong or long-term hypoxia should be avoided.

RESPONSE: There is no solid definition for the upper limit of hypoxia, but we used 2.8 mg $O_2$/L (~2.0 ml $O_2$/L) as the upper limit, which is defined in some influential papers and has been widely applied (Wu, 2002; Diaz and Rosenberg, 2008). Therefore, 2.0 mg $O_2$/L used in our study is definitely low enough to be considered hypoxic according to the conventional standard. We can mention that the results indicate the impacts of short-term hypoxia on calcification.

Diaz, R.J., and Rosenberg, R.: Spreading dead zones and consequences for marine ecosystems. Science, 321, 926–929, 2008.

Wu, R.S.S.: Hypoxia: from molecular responses to ecosystem responses. Mar. Pollut. Bull., 45, 35–45, 2002.

6) In lines 96-97 authors say that part of the individuals was left into hypoxic conditions during one week before the experiment (after damage and measures of the tubes). Can you please justify this choice? Does this mean that the T0 for tube sizes represents instead one week under treatment conditions? Did the authors measure the T0 size again after the acclimation and before the start of the experiment?

RESPONSE: This acclimation period is needed to remove the fight-or-flight response after shell breaking. The approximate tube length after shell breaking is provided (Ln 93), but it does not represent the initial tube length. We measured the tube length on Day 1 of the exposure period as the initial tube length (Ln 100).

7) How did the authors get to have stable oxygen conditions during and just after water changes (every 3 days)? Did you measure oxygen into the culture solution before retiring old seawater? Could you please show these data somewhere in supplementary materials?

RESPONSE: Stable DO concentration at hypoxic level can be obtained by aerating the seawater continuously with a mixture of nitrogen and air (Ln 105-107), where the flow rates of these two gases were adjusted to achieve the target DO level (Ln 85-88). Stable equilibrium between gases in seawater can be achieved by this method quickly so that the DO concentration in seawater can be very stable throughout the 3-week exposure period. This method has been widely used for hypoxia study (e.g. Leung et al., 2013; Mukherjee et al., 2013). The DO concentration of seawater was measured daily.

Leung, J.Y.S., Cheung, S.G., Qiu, J.W., Ang, P.O., Chiu, J.M.Y., Thiyagarajan, V., and Shin, P.K.S.: Effect of parental hypoxic exposure on embryonic development of the offspring of two serpulid polychaetes: Implication for transgenerational epigenetic effect. Mar. Pollut. Bull., 74, 149–155, 2013.

Mukherjee, J., Wong, K.K.W., Chandramouli, K.H., Qian, P.Y., Leung, P.T.Y., Wu, R.S.S., and Thiyagarajan, V.: Proteomic response of marine invertebrate larvae to ocean acidification and hypoxia during metamorphosis and calcification. J. Exp. Biol., 213, 4580–4589, 2013.

8) Is edx analysis resolution enough for consistent Mg/Ca measurement in the shell? Can the authors give more information about the accuracy, precision, detection limits etc. of the analysis, please? Or add references if the protocol is routinely used.

RESPONSE: EDX (or EDS) has been extensively used to analyse Mg/Ca in the shell with good results. We can add references in the text (e.g. Ries, 2004; Zhang et al., 2010).

Ries, J.B.: Effect of ambient Mg/Ca ratio on Mg fractionation in calcareous marine invertebrates: A record of the oceanic Mg/Ca ratio over the Phanerozoic. Geology, 32, 981–984, 2004.

Zhang, F., Xu, H., Konishi, H., Roden, E.E.: A relationship between $d_{104}$ value and composition in the calcite-disordered dolomite solid-solution series. Am. Mineral., 95, 1630–1656, 2010.

9) Some of the conclusions/discussions are inferential and not supported by the data. For example paragraph from line 249 to 258 should be deleted, in my opinion, as it is not supported by the presented data.

RESPONSE: This paragraph is a general discussion on the mechanism affecting calcification and our results (e.g. seawater carbonate chemistry and shell growth) can support this discussion. This paragraph is important to reshape the concept of calcification and to improve the readership of our manuscript.

10) The hypothesis of relaxed magnesium regulation to explain higher Mg/Ca in the calcite produced under hypoxia is based on benthic foraminifera. These organisms are known to strongly discriminate against magnesium. This does not seem the case for polychaetes, which seem to contain very high concentrations of Mg in the shell. Authors should base their hypotheses on more adapted literature.

RESPONSE: We cannot validate this comment. The Mg/Ca in calcite varies greatly among foraminifera species, where many of them have Mg/Ca greater than 15 mol %, such as *Spiroloculina clara*, *Planispirinella exigua*, *Peneroplis proteus*, *Alveolinella quoii* and *Mychostomina revertens* (Blackmon and Todd, 1959). Please also note that the transport and sequestration of Mg are heavily regulated in eukaryotic cells and the underlying mechanisms are likely conserved among eukaryotic species. Thus, it is reasonable to conjecture that similar mechanism can be shown in our tested calcifying polychaete. In fact, the currently acknowledged regulatory mechanism of Mg in foraminifera is partially inferred from the combined biological knowledge on *Paramecium*, mollusks and cultured rodent cells (see Bentov and Erez, 2006).

Blackmon, P.D., and Todd, R.: Mineralogy of some foraminifera as related to their classification and ecology. J. Paleontol., 33, 1–15, 1959.

Bentov, S. and Erez, J.: Impact of biomineralization processes on the Mg content of foraminiferal shells: A biological perspective. Geochem. Geophys. Geosyst., 7, Q01P08, 2006.

**Minor comments/suggestions:**

Line 30: use "increase" instead of "augment"

RESPONSE: Suggestion will be adopted.

Line 51: replace "It" by "This"

RESPONSE: Suggestion will be adopted.

Line 53: What does "or other physiological processes via energy trade-off" mean? Can you explain what other processes you're talking about please?

RESPONSE: Based on the energy budget model, other processes can include growth, reproduction, somatic maintenance, etc. We will revise this sentence for clarity.

Line 58: Please delete "and defense response".

RESPONSE: "Defense response" has a specific meaning in this study, which refers to the response following non-lethal shell damage.

Materials and methods: Please add titration protocol for alkalinity measures presented in table S1.

RESPONSE: It is unnecessary to add a protocol for alkalinity measurement because it is far too technical and general readers are not interested in how to operate a particular model of titrator. We have never seen research articles, except method papers, describing the operating procedures for a particular model of equipment. If the reviewer feels interested in the titration protocol for this model, here is the user manual: https://www.manualslib.com/manual/530078/Hanna-Instruments-Hi-84431.html#manual

Lines 93-97: Please add more details about the procedure used to measure the specimens, associated errors and discuss the potential stress that this manipulation may represent and the effect it could have on the final results

RESPONSE: We can add these details in the revision. However, please note that instant stress due to fight-or-flight response would be induced when breaking the tube. To reduce this stress, the individuals were allowed to rest for a week prior to experimentation (Ln 94) and therefore they were only under stress due to non-lethal shell damage, which is a factor in our experiment.

The order of paragraphs in the Material and methods section is, at present, a bit confusing and should follow a more logical pattern. I would suggest that the paragraph on experimental design should show the setting, the replication, times etc., for all type of analyses. Then a paragraph on procedures for physiological analyses should explain all the used methods for respiration rates, survival rates, feeding rates and shell growth. Then a final paragraph on the shell composition should follow.

RESPONSE: We appreciate reviewer's suggestions and will change the order accordingly.

Lines 99-100: Why is the color of new shell very different from the ancient one? Is it normal? Maybe you should discuss it somewhere. One would think that it is a culturing effect on shell structure... Figure S1 should be in the main text.

RESPONSE: The colour of new shells should be white due to the presence of calcium carbonate. Yet, the colour will turn slightly yellowish over time because of the biofilm (e.g. bacteria, algae, etc.) growing on the surface. It is very normal and important because the biofilm allows polychaete larvae to settle and form a colony. We can put Figure S1 in the main text.

Lines 103-112 should be part of the "Experimental set up" paragraph

RESPONSE: We will change the order of paragraphs as suggested above.

Line 123: The specimens used for organic matter composition are the same measured for toughness? Please detail this kind of information

RESPONSE: It cannot be, unfortunately, because removing the organic matter in the shell at 550°C can substantially affect shell toughness. We will revise the sentence for clarity.

Paragraph 123-129: 1) for the composite shell power did you mix calcite from different specimens, isn't it? Did they come from a same replicate for a treatment or even replicates were mixed? How did you prepare the powder exactly? Did you wash the shell before to avoid contamination (from algae given as food for example)? How? Why did you choose Mg/Ca ratio as a parameter to be measured?

RESPONSE: For each composite sample, shells from 3-5 individuals in the same replicate were used (Ln 124) so that we had 3 replicates per treatment for the geochemical properties. The powder was prepared by removing the newly-produced shells, rinsing them with deionized water (to remove the microalgae and other debris), drying them at room temperature and finally grinding them using a mortar and pestle. Many calcifying organisms can change Mg/Ca in their calcitic shells in response to the changing environment (e.g. ocean acidification). The underlying mechanism remains largely unknown, but may be associated with metabolic energy that can be greatly affected by hypoxia. Therefore, we expected that Mg/Ca would change in response to hypoxia.

Paragraph 148-155: How did you calibrate oxygen probes before the analysis?

RESPONSE: Before the analysis, the probe was calibrated by inputting the value of calibration slope in the software. Then, we have to validate the DO concentration using another DO meter (e.g. handheld DO meter), which is calibrated by measuring the DO concentration of oxygen-saturated seawater. We

may need to change the value of calibration slope until the DO concentrations between DO probes are the same. For the operation of the instrument, please refer to the user manual: https://in-situ.com/wp-content/uploads/2015/05/Stable_Optical_Oxygen_System_-SOO-100_Manual.pdf

Line 152: "The air inside the syringe...": what air? Why is it there during the first measure? This part is not clear to me.
RESPONSE: The air is atmospheric air. For the initial measurement, the individuals were allowed to rest in the syringe for 15 min and the small pocket of air can help buffer the change in DO concentration during this period. We can add this information for clarity.

Line 154: "by gently stirring the FSW inside the syringe". How did you avoid oxygenation at this step?
RESPONSE: Such gently stirring can only help homogenize the DO concentration in the water body. Oxygenation of water is caused by dissolving atmospheric oxygen in water. However, this process cannot occur because atmospheric air cannot interact with the water inside the syringe, which is under air-tight conditions (Ln 151-152).

Line 156: Please specify the unit used for consumed oxygen. Normally mL or μmol are used. In your figure 4a you use μg $O_2$, which is quite unusual as a unit for oxygen.
RESPONSE: For the unit of consumed oxygen, mg $O_2$ is frequently used, while μg $O_2$ is just a conversion for the magnitude to avoid many decimal places.

Lines 159-160: Why did you only used one algal species for this experiment? Can you add fundamental details such as if the experiment was performed under light conditions, please? Also you say that 5 replicated bottles were used per treatment. In the first materials and methods section you say you have 3 replicated bottles with 10 specimens for the experiments. I'm lost... Are these different bottles? How many individuals per bottle per treatment do you have then?
Statistical paragraph: Where data transformed before the analyses to homogenize the magnitudes? How many permutations were performed? Which distance parameter was used and why? Can you specify the "aforementioned parameters" of line 168, please?
RESPONSE: One algal species should be used for the feeding experiment to avoid the selective feeding of filter feeders (due to different sizes and/or textures between microalgae), which can complicate the calculation and interpretation. We can add a sentence to state that the experiment was performed under light conditions. We used different bottles for the feeding and shell growth experiments (5 replicate bottles with 5 individuals per bottle for the feeding experiment, Ln 156-158; 3 replicates bottles with 10 individuals per bottle for the shell growth experiment, Ln 102 and 105). For univariate analysis, data are on the same scale (i.e. carrying the same unit), therefore transformation is unnecessary. The number of permutation is 999 and Euclidean distance was used which is commonly applied. We will add these details in the revision. We will also list the parameters (e.g. respiration rate, clearance rate, shell toughness, etc.) in the revision.

Line 172: You say hypoxia slightly hindered, but your statistics say this difference is significant, so maybe this should be emphasized a bit more.
RESPONSE: Please note that "statistically significant" does not necessarily imply "biologically significant". We can emphasize the statistical significance a bit more, but it is more important to compare the two factors with respect to the magnitude of change, which is of biological importance.

Line 174: Please replace "negligible" with "no".

RESPONSE: "no" is too absolute. We prefer to be conservative by using "no observable" or "no significant".

Line 180: You say "but only slightly by non-lethal shell damage". It looks like a significant difference, visually (fig. 4a). Is it confirmed by statistical analyses? Yes, so you should say it!
RESPONSE: It is statistically significant (Table S2). We can revise this sentence in a more explicit way.

Lines 181-182: specify whether statistical differences are visible and where.
RESPONSE: We can specify this in the revision.

Line 187: what do you mean by "ramifications"?
RESPONSE: It can mean changes in species populations, community structure and ecosystem functioning. We can make the meaning more explicit.

Line 188: "tolerant to hypoxia", at what temporal scale?
RESPONSE: Short-term scale.

Line 194: I would suggest not to use "unthreatened conditions" as a general for "shell damage", because hypoxia also is an unthreatened condition, so it can result confusing.
RESPONSE: We appreciate this suggestion, but we have explicitly defined "unthreatened conditions" as "individuals without shell damage" in the methods and mentioned the definition again in this sentence. There should be no confusion.

Line 195: "hypoxia slightly hinders"... again, is it significant or not?
RESPONSE: Significant (see Table S2).

Lines 201-202: redundant concept, already said.
RESPONSE: We can delete this sentence.

Lines 205-206: Please replace "shell growth" with "inorganic components of the shell". You suggest that under hypoxia the production of organic matter compensate for diminished quality of inorganic components (>ACC). Palmer (1992) on the contrary suggests that organic matter production is costly and that would be the reason why high-organic calcareous microstructures became rare with evolution. How do you explain this incoherence?
RESPONSE: Suggestion for the word choice can be adopted. However, we never suggest that "under hypoxia the production of organic matter compensate for diminished quality of inorganic components (>ACC)". In this paragraph, we discussed the effects of hypoxia on shell growth and shell strength under unthreatened conditions. As the organic matter content was not affected by hypoxia, we suggest that the polychaetes still allocate similar amount of energy to maintain shell strength at the expense of shell growth under hypoxia.

Line 211-212: Could the overproduction of organic matter be related to higher calcification rates?
RESPONSE: Although we cannot rule out this possibility, we think it is still premature to make this speculation because we found that organic matter content does not necessarily increase with calcification rate (Normoxia vs. Hypoxia without shell damage).

Lines 219-221: I do not understand what you mean

RESPONSE: This sentence means when the individual is under life-threatening conditions and chance of survival becomes very low, it has to prioritize defense response as the last resort to maximize survival rate. We will rephrase this sentence to better illustrate the idea.

Line 224: You say the effect of hypoxia on defense response is not discernible. Although toughness is not affected, I would say that reduced shell growth rates (visible in figure 1) should be taken into account, to discern the effects of hypoxia, as well.
RESPONSE: We will revise this sentence by considering the effect of hypoxia on shell growth.

Line 234: please replace "signifies" with "may suggest".
RESPONSE: Suggestion will be adopted.

Line 241: At the end of the sentence you should add ".. can generally be maintained at least under slight hypoxia, on a short timescale."
RESPONSE: Suggestion will be adopted.

Line 248: please delete "and therefore *H. diramphus* prioritized defense response".
RESPONSE: Suggestion will be adopted.

Line 259: please add "open" between marine and waters (because "costal" are also marine waters!)
RESPONSE: Suggestion will be adopted.

Line 264: when you say that the defense response can be sustained you should specify "on a short timescale", because your results are based on short-term experiments.
RESPONSE: We appreciate this suggestion and will specify this in the text.

Lines 330-339: Please order the references on a chronological basis.
RESPONSE: Suggestion will be adopted.

Line 338 and 368: add authors to the list
RESPONSE: We will revise them according to the journal's latest style.

Survival rate figure (S2) should not be in supplementary material, as this is an important result to take into account in the analysis of all the others.
RESPONSE: We will put this figure in the main text.

---

## Author Comment (AC3) · 15 Dec 2017

Dear Editor,

The manuscript "Calcification and inducible defense response of a calcifying organism could be maintained under hypoxia through phenotypic plasticity" by Leung and Cheung raises a few interesting questions regarding organismal adaptations to hypoxia. Although authors present a few interesting ideas, I do not find the manuscript in its current form suitable for publication due to a significant shortfall in methods description, exaggerated interpretation of results and an incoherent discussion. However, if the Discussion paper manuscript progresses for publication in Biogeosciences, I suggest that the authors consider some major revisions that I have described in detail in the supplement. Please find attached my suggestions for major and minor revisions.

RESPONSE: We thank this reviewer for reviewing and giving suggestions for our manuscript. We found that most of the comments are related to the methods and can be clarified easily. The only two comments for the discussion section are minor and can be easily addressed as well. Therefore, we are happy to incorporate the good suggestions into the revision.

Major Comments:

Title: I do not find this title representative of the authors results/discussion. Please describe which aspects of the phenotype are considered plastic, since there is no change in mechanical strength and authors discuss less crystallographic control and magnesium regulation under hypoxic conditions.

RESPONSE: The plastic traits are the mineralogical properties of shells. We will make the title more descriptive in the revision.

Lines 34-38: Authors seem to go around a point here. Please state exactly what drives calcification instead of pH and seawater chemistry.

RESPONSE: The proposed main driver for calcification is the energetics of organisms (Ln 38-39, 41-42), which can be substantially affected by hypoxia (Ln 39-43).

Line 54: Energy costs can increase rather than reduce, if organic content is modified such that it increases.

RESPONSE: The word "modify" does not indicate the direction of change. We will change it to "decrease" to avoid ambiguity.

Line 76: Mean size of polychaetes?

RESPONSE: The tube length of polychaetes mostly ranged from 35 to 45 mm. This will be added to the revision.

Line 102: Diameter of hole drilled?

RESPONSE: About 2 mm. This will be added to the revision.

Line 103: Tubes were glued using what?

RESPONSE: Hot-melt adhesive. This will be added to the revision.

Line 114: How was the shell fragment acquired and cleaned of organic tissue?

RESPONSE: Shell fragments were obtained by breaking the newly-produced shells using a pair of forceps and they were then cleaned by rinsing with deionized water. Please note that the flesh of polychaetes does not attach to the shell. This information will be added to the revision.

Line 116: Was the same surface (e.g: inner shell surface) always used for indentation?

RESPONSE: Yes, we always used the inner shell surface for indentation. This will be clarified in the revision.

Line 122: Organic content of shell or whole polychaete? It is of the shell I assume, but is unclear.

RESPONSE: Only the newly-produced shells were used for analysis of organic matter content. We will make this sentence clearer.

Line 126: How was the shell powder acquired?

RESPONSE: Shell powder was obtained by grinding the newly-produced shells using a mortar and pestle. This will be clarified in the revision.

Line 130: Please provide information on how these polymorphs are typically distributed in the organism. Are the polymorphs specific to the outer/inner layer of the shell?

RESPONSE: We do not have information on the distribution of these carbonate polymorphs. Yet, it is not necessary to know that because it is irrelevant to our research question.

Line 139: Were ACC, aragonite and calcite standards measured? Please explain why, if not.

RESPONSE: Standards are used for determination of absolute quantity, but only relative quantity of these parameters is needed in our study. Specifically, since relative ACC content is indicated by the peak ratio in the IR spectrum, it is not necessary to measure the standard as long as background calibration for the baseline is made (Chan et al., 2012; Leung et al., 2017). As for the calcite to aragonite ratio, we apply the calibration equation in a method paper (Kontoyannis and Vagenas, 2000), which is derived by using pure calcite and aragonite.

Chan, V.B.S., Li, C., Lane, A.C., Wang, Y., Lu, X., Shih, K., Zhang, T. and Thiyagarajan, V: $CO_2$-driven ocean acidification alters and weakens integrity of the calcareous tubes produced by the serpulid tubeworm, *Hydroides elegans*. PLoS ONE, 7, e42718, 2012.

Leung, J.Y.S., Connell, S.D., Nagelkerken, I. and Russell, B.D.: Impacts of near-future ocean acidification and warming on the shell mechanical and geochemical properties of gastropods from intertidal to subtidal zones. Environ. Sci. Technol., 51, 12097–12103, 2017.

Kontoyannis, C.G. and Vagenas, N.V.: Calcium carbonate phase analysis using XRD and FT-Raman spectroscopy. Analyst, 125, 251−255, 2000.

Line 140: Diameter of the KBR-shell powder disc?

RESPONSE: About 2 cm. This information will be added to the revision.

FTIR: FTIR is a bulk measurement and ideally should not be used to infer "relative" proportions of carbonate polymorphs. Typically, the presence of a 713 cm-1 peak is indicative of crystalline calcium carbonate comprising the bulk of shell carbonates. However, I am aware that this interpretation has been used before and if authors proceed with the analyses, could they please clarify if the spectra were scaled so that 713 cm-1 peaks had the same heights as described in Weiss et al (2002)? In addition, please specify the typical size of crystallites in shell since such ratios have been demonstrated to be influenced by particle size (Kristova et al 2015).

RESPONSE: FTIR has been widely used to indicate the relative ACC content by measuring the peak ratio between 856 cm$^{-1}$ and 713 cm$^{-1}$ (e.g. Beniash et al., 1997; Chan et al., 2012; Leung et al., 2017). Since relative ACC content is indicated by this ratio rather than an absolute peak height, it is not necessary to rescale the peak height at 713 cm$^{-1}$ to that in Weiss et al. (2002). The particle size of shell powder was ~5 µm.

Beniash, E., Aizenberg, J., Addadi, L., and Weiner, S.: Amorphous calcium carbonate transforms into calcite during sea urchin larval spicule growth. Proc. R. Soc. B, 264, 461–465, 1997.

Line 148: What were the syringes made to of?

RESPONSE: Polypropylene plastic. We will add this information in the revision.

Line 156: Hunger is only standardised if individuals were at the same start point.

RESPONSE: Therefore, all individuals were starved for 1 day prior to feeding trials. This is more than enough for them to clear their gut content.

Lines156-166: This doesn't represent clearance rates during the experiment.

RESPONSE: We disagree. This clearance method has been widely applied for determination of clearance/filtering/feeding rate of feeding feeders (e.g. Riisgård, 2001; Contreras et al., 2012; Leung et al., 2013; Leung and Cheung, 2017).

Contreras, A.M., Marsden, I.D., and Munro, M.H.G.: Effects of short-term exposure to paralytic shellfish toxins on clearance rates and toxin uptake in five species of New Zealand bivalve. Mar. Freshw. Res., 63, 166–174, 2012.

Leung, J.Y.S. and Cheung, N.K.M.: Feeding behaviour of a serpulid polychaete: Turning a nuisance species into a natural resource to counter algal blooms? Mar. Pollut. Bull., 115, 379–382, 2017.

Leung, Y.S., Shin, P.K.S., Qiu, J.W., Ang, P.O., Chiu, J.M.Y., Thiyagarajan, V., and Cheung, S.G.: Physiological and behavioural responses of different life stages of a serpulid polychaete to hypoxia. Mar. Ecol. Prog. Ser., 477, 135–145, 2013.

Riisgård, H.U.: On measurement of filtration rates in bivalves — the stony road to reliable data: review and interpretation. Mar. Ecol. Prog. Ser., 211, 275–291, 2001.

Line 170-180: Please provide full FTIR spectra as a supplementary figure.

RESPONSE: We can provide a FTIR spectrum as a supplementary figure.

Lines 267-268: Can inferences be made regarding whether inner/outer layers were calcified if the polymorphs are specific to a layer of the polychaete shell?

RESPONSE: We cannot make this inference based on our results. As mentioned above, further investigation on structural properties is needed to answer this question, which is beyond the scope of this study.

Line 233-234: This is a strong statement. Regulation of Mg may be interpreted but the authors results do not signify that it is relaxed under hypoxia.

RESPONSE: We will tone down this statement, as suggested by another reviewer.

Line 256-257: Please delete this final sentence. It is a very strong statement and the whole paragraph does not explain why hypoxia the key stressor in the future (which is debatable anyway).

RESPONSE: We will delete this sentence.

Minor Comments:

Line 10-11: Sentence like this needs a reference.

RESPONSE: In the Abstract, citations should be avoided.

Line 25: change "shells" to skeletons.

RESPONSE: We will replace "shells" with "shells or skeletons".

Line 32: Delete "however".

RESPONSE: Suggestion will be adopted.

Line 368: Please provide full reference.

RESPONSE: We will follow the journal's latest style.

Figure 5 (SEM): Are these images of the aragonite or calcitic parts of the shell? The legend needs more descriptive text. It is not obvious to me how these images indicate shell integrity.

RESPONSE: We cannot identify the type of carbonate mineral based on these images and this is beyond the scope of this imaging analysis. We will elaborate the figure legend to indicate shell integrity in terms of crystal thickness and density.

Table A1: Please include other calculated parameters such as HCO3-, CO32- and CT.

RESPONSE: Suggestion will be adopted.

References used for review:

Weiss et al (2002) Mollusc larval shell formation: amorphous calcium carbonate is a precursor phase for aragonite. DOI: 10.1002/jez.90004

Kristova et al (2015) The effect of the particle size on the fundamental vibrations of the [CO3(2-)] anion in calcite. DOI: 10.1021/acs.jpca.5b02942.

---

## Author Response (AR1)

22 March 2018

Dear Prof. Treude,

Please consider our REVISED manuscript entitled "Effects of hypoxia and non-lethal shell damage on shell mechanical and geochemical properties of a calcifying polychaete" for publication as a research article in *Biogeosciences*.

We appreciate the anonymous reviewers for their suggestions to improve the impact of our manuscript. We have incorporated their suggestions into the manuscript and believe that the problems in the Materials and Methods have been solved. For those suggestions which are incorrect or obviously beyond the scope of study, we have fully explained in the Response. The changes have been stated in the Response with the line number and shown in the highlighted version of revised manuscript.

We would deeply appreciate an opportunity to respond to further review comments. Thank you very much for considering our manuscript again and we look forward to your positive reply at your earliest convenience.

Yours sincerely,

Dr. Jonathan Leung
Corresponding author
On behalf of Napo Cheung

**Reviewer 1**

Dear Editor,

The manuscript "Calcification and inducible defense response of a calcifying organism could be maintained under hypoxia through phenotypic plasticity" by Leung and Cheung, presents interesting questions about possible eco-physiological adaptations observed on a calcifying polychaete exposed to acute hypoxia, including changes in calcification rates, shell composition and metabolism.

Despite some interesting points, I think that the manuscript, in the present form is not acceptable for publication, because of a substantial lack of detail on the protocols used for the experiments and the analyses, very shallow description of the main results and some over-interpretation of the results. A (very) major revision is therefore suggested.

RESPONSE: We thank this reviewer for reviewing and providing useful suggestions for improving our manuscript. After reading the comments carefully, we found that most of them are related to the methods and can be clarified easily. However, there are some invalid arguments which are explained below. We have prepared a major revision of our manuscript by incorporating the good suggestions.

Here listed some of the **major comments**:

1) The materials and methods are too undetailed. Are the specimens at the T0 adult or juveniles? Were they exposed to a day/night light cycle? If so, how did you control algal proliferation during the experiment? If not, are you sure that this does not influence their physiology? Were added algae dead or alive? What basis the 20 ml algal concentration was chosen on? Before organic matter analyses, were the samples washed? If so, how? Also for statistical analyses more details are needed.

RESPONSE: The specimens were adults (Ln 78) and exposed to a day/light cycle of 14:10 hrs (Ln 115). Live algae were added (Ln 115), but there was no need to control their growth because the consumption rate of polychaetes is faster than the proliferation rate of algae. That is why we had to add algal suspension on a daily basis to sustain the growth of polychaetes (Ln 116). The feeding regime was based on our extensive experience in rearing *Hydroides* spp., which can ensure their normal growth under laboratory conditions. The shells were rinsed with deionized water before organic matter analysis (Ln 160-161). The above information has now been added in the revision and more details about statistical analysis have been provided (Ln 199-202).

2) I do not understand why in the materials and methods the authors say that they used 10 specimens per replicate (x 3 repl.) per treatment, but each observed response in the results is based on only n=3 or n=5 specimens (or replicates??). This is surprising if we consider that most of the performed analyses (e.g. respiration rates, growth rates, clearance rate) are not destructive and that just part of the shell is necessary for the rest of the analyses (e.g. Mg/Ca, organic matter, ACC...). So why did not the author use more than 3 or 5 specimens to do their analyses and calculate averages? Could they add more replicates?

RESPONSE: We adhere to the classic, unambiguous definition of "n" as "number of replicates" (i.e. not necessarily "number of individuals"). We have carefully clarified the exact sample size used for each of the measurements in our revised manuscript.

   We initially had a total of 30 individuals per treatment (Ln 103). Since tube growth can be measured with sufficient accuracy and precision (Ln 128), the tube growth of each individual was analysed as one replicate, hence resulting in a sample size of 30 (Ln 125). The same concept applies to the measurements of organic matter and shell toughness, where accurate measurements could be made on individual shell (i.e. one shell from one individual as a replicate). However, due to the small size of the polychaetes and the limited quantity of the newly-formed shells, multiple individuals or their newlyformed shells were pooled as composite samples for the remaining assays. Each of these composite samples (instead of the underlying individuals) was treated as a single statistical entity, i.e. a replicate. Based on our experience and preliminary test, this design provides adequate precision of measurement and statistical power. This is also evident from the small standard errors in our results (see Fig. 4 and 5).

Please note that although the respiratory rate and clearance rate were reported with a unit of "$ind^{-1}$ $hr^{-1}$", this is merely a mathematical expression. For these measurements, 25 individuals were made into 5 composites (replicates) with 5 individuals per replicate. The measured rates reported for each replicate was simply expressed as an average of the pooled individuals (the units of these variables were expressed as "per individual"), while the effective sample size for the statistical analyses remains as "5 (composites)" instead of "25 (individuals)".

3) The experimental design is quite weak. Maybe this impression derives also from the difficulty to understand how many individuals/measures/replicates where performed for each parameter. We lack fundamental information about the ability of these organisms to develop into cultures. All the results are discussed on the basis of comparison to normoxic conditions. How can the authors be sure that a "culturing effect" is not interfering with the results? This is particularly true for shell composition. Why did not the author compare the shells of individuals grown in normoxic conditions to pre-experiment portions of shells (grown into natural conditions) to see if any "culturing effect" is visible?

RESPONSE: We have now revised the experimental design section to clearly define replicates. As stated in the Introduction, this study aims to elucidate the effects of hypoxia on calcification and defence response of a calcifying polychaete, which can be experimentally tested under fully controlled laboratory conditions in order to isolate the effects of hypoxia and minimize the number of confounding factors. In contrast, the "culturing effect" cannot answer our research question because it only indicates the effects caused by the difference between laboratory and field conditions, but not the effects of hypoxia. As all the individuals were reared under laboratory conditions, the results were not influenced (or biased) by "culturing effect".

4) The protocol used for respiration measurements is unusual to me. I cannot understand how the authors did measure the oxygen content of hypoxic waters into a syringe with a relatively thick probe tip and can be sure they avoided oxygenation during the measurement. Also, are they sure that the material the syringe is made of is impermeable to oxygen? Do they have any measure of blanks to estimate the possible gas exchange through the syringe walls during the analysis?

RESPONSE: We have to invalidate this comment. Our DO probe is thin (~2.4 mm in diameter) enough to perfectly insert into the hole of plastic syringe (~2.7 mm in diameter) for DO measurement. The syringes used were ordinary plastic syringes (Terumo Hypodermic Syringe without a needle) with a barrel wall which is definitely thick enough to be impermeable to air. Once the syringe is sealed (Ln 141), oxygenation cannot occur under such air-tight conditions. This simple method has been widely used before (e.g. Zhao et al., 2011; Leung et al., 2013).

Leung, Y.S., Shin, P.K.S., Qiu, J.W., Ang, P.O., Chiu, J.M.Y., Thiyagarajan, V., and Cheung, S.G.: Physiological and behavioural responses of different life stages of a serpulid polychaete to hypoxia. Mar. Ecol. Prog. Ser., 477, 135–145, 2013.

Zhao, Q., Cheung, S.G., Shin, P.K.S., and Chiu, J.M.Y.: Effects of starvation on the physiology and foraging behaviour of two subtidal nassariid scavengers. J. Exp. Mar. Biol. Ecol., 409, 53–61, 2011.

5) In the experiment the authors consider hypoxic conditions to be reached at ~2.0 mg O2/L. This concentration is the one which is normally considered to be the upper limit for hypoxia. Considering that the error associated to the probe used for $O_2$ survey is 0.1 mg/L (source: TauTheta manual) and that the average values the authors report in table S1 are always a slightly above 2.0 mg/L for hypoxic conditions, I wonder why the authors did not test a lower concentration to be sure to never exceed oxygen concentrations corresponding to namely hypoxic conditions. The results obtained should be carefully presented as the response to a case of acute (short time) and slight (upper limit) hypoxia and every over-interpretation or generalization to strong or long-term hypoxia should be avoided.

RESPONSE: There is no solid definition for the upper limit of hypoxia, but we used 2.8 mg $O_2$/L (~2.0 ml $O_2$/L) as the upper limit, which is defined in some influential papers and has been widely applied (Wu, 2002; Diaz and Rosenberg, 2008). Therefore, 2.0 mg $O_2$/L used in our study is definitely low enough to be considered hypoxic according to the conventional standard. We have also mentioned in the revision that the results indicate the impacts of short-term hypoxia on calcification.
Diaz, R.J., and Rosenberg, R.: Spreading dead zones and consequences for marine ecosystems. Science, 321, 926–929, 2008.
Wu, R.S.S.: Hypoxia: from molecular responses to ecosystem responses. Mar. Pollut. Bull., 45, 35–45, 2002.

6) In lines 96-97 authors say that part of the individuals was left into hypoxic conditions during one week before the experiment (after damage and measures of the tubes). Can you please justify this choice? Does this mean that the T0 for tube sizes represents instead one week under treatment conditions? Did the authors measure the T0 size again after the acclimation and before the start of the experiment?

RESPONSE: This acclimation period is needed to remove the fight-or-flight response after shell breaking (Ln 99-101). The approximate tube length after shell breaking is provided (Ln 98), but it does not represent the initial tube length. We measured the tube length on Day 1 of the exposure period as the initial tube length (Ln 104, 125).

7) How did the authors get to have stable oxygen conditions during and just after water changes (every 3 days)? Did you measure oxygen into the culture solution before retiring old seawater? Could you please show these data somewhere in supplementary materials?

RESPONSE: Stable DO concentration at hypoxic level can be obtained by aerating the seawater continuously with a mixture of nitrogen and air, where the flow rates of these two gases were adjusted to achieve the target DO level (Ln 92-95). Stable equilibrium between gases in seawater can be achieved by this method quickly (< 5 min) so that the DO concentration in seawater can be very stable throughout the 3-week exposure period (Ln 109-113). This method has been widely used for hypoxia study (e.g. Leung et al., 2013; Mukherjee et al., 2013). The DO concentration data have been added in Figure A1.
Leung, J.Y.S., Cheung, S.G., Qiu, J.W., Ang, P.O., Chiu, J.M.Y., Thiyagarajan, V., and Shin, P.K.S.: Effect of parental hypoxic exposure on embryonic development of the offspring of two serpulid polychaetes: Implication for transgenerational epigenetic effect. Mar. Pollut. Bull., 74, 149–155, 2013.
Mukherjee, J., Wong, K.K.W., Chandramouli, K.H., Qian, P.Y., Leung, P.T.Y., Wu, R.S.S., and Thiyagarajan, V.: Proteomic response of marine invertebrate larvae to ocean acidification and hypoxia during metamorphosis and calcification. J. Exp. Biol., 213, 4580–4589, 2013.

8) Is edx analysis resolution enough for consistent Mg/Ca measurement in the shell? Can the authors give more information about the accuracy, precision, detection limits etc. of the analysis, please? Or add references if the protocol is routinely used.

RESPONSE: EDX (or EDS) has been extensively used to analyse Mg/Ca in the shell with good results. We have added references in the text (e.g. Ries, 2004; Zhang et al., 2010; Leung et al., 2017) (Ln 187).

Ries, J.B.: Effect of ambient Mg/Ca ratio on Mg fractionation in calcareous marine invertebrates: A record of the oceanic Mg/Ca ratio over the Phanerozoic. Geology, 32, 981–984, 2004.

Zhang, F., Xu, H., Konishi, H., Roden, E.E.: A relationship between $d_{104}$ value and composition in the calcite-disordered dolomite solid-solution series. Am. Mineral., 95, 1630–1656, 2010.

Leung, J.Y.S., Connell, S.D., Nagelkerken, I., and Russell, B.D.: Impacts of near-future ocean acidification and warming on the shell mechanical and geochemical properties of gastropods from intertidal to subtidal zones. Environ. Sci. Technol., 51, 12097–12103, 2017.

9) Some of the conclusions/discussions are inferential and not supported by the data. For example paragraph from line 249 to 258 should be deleted, in my opinion, as it is not supported by the presented data.

RESPONSE: This paragraph is a general discussion on the mechanism affecting calcification and our results (e.g. seawater carbonate chemistry and shell growth) can support this discussion. This paragraph is important to reshape the concept of calcification and to improve the readership of our manuscript.

10) The hypothesis of relaxed magnesium regulation to explain higher Mg/Ca in the calcite produced under hypoxia is based on benthic foraminifera. These organisms are known to strongly discriminate against magnesium. This does not seem the case for polychaetes, which seem to contain very high concentrations of Mg in the shell. Authors should base their hypotheses on more adapted literature.

RESPONSE: We cannot validate this comment. The Mg/Ca in calcite varies greatly among foraminifera species, where many of them have Mg/Ca greater than 15 mol %, such as *Spiroloculina clara*, *Planispirinella exigua*, *Peneroplis proteus*, *Alveolinella quoii* and *Mychostomina revertens* (Blackmon and Todd, 1959). Please also note that the transport and sequestration of Mg are heavily regulated in eukaryotic cells and the underlying mechanisms are likely conserved among eukaryotic species. Thus, it is reasonable to conjecture that similar mechanism can be shown in our tested calcifying polychaete. In fact, the currently acknowledged regulatory mechanism of Mg in foraminifera is partially inferred from the combined biological knowledge on *Paramecium*, mollusks and cultured rodent cells (see Bentov and Erez, 2006).

Blackmon, P.D., and Todd, R.: Mineralogy of some foraminifera as related to their classification and ecology. J. Paleontol., 33, 1–15, 1959.

Bentov, S. and Erez, J.: Impact of biomineralization processes on the Mg content of foraminiferal shells: A biological perspective. Geochem. Geophys. Geosyst., 7, Q01P08, 2006.

**Minor comments/suggestions:**

Line 30: use "increase" instead of "augment"
RESPONSE: Suggestion adopted (Ln 30).

Line 51: replace "It" by "This"
RESPONSE: Suggestion adopted (Ln 52).

Line 53: What does "or other physiological processes via energy trade-off" mean? Can you explain what other processes you're talking about please?
RESPONSE: Based on the energy budget model, other processes can include growth, reproduction, somatic maintenance, etc (Ln 54-55).

Line 58: Please delete "and defense response".

RESPONSE: "Defense response" has a specific meaning in this study, which refers to the response following non-lethal shell damage.

Materials and methods: Please add titration protocol for alkalinity measures presented in table S1.

RESPONSE: It is unnecessary to add a protocol for alkalinity measurement in the manuscript because it is far too technical and general readers are not interested in how to operate a particular model of titrator. We have never seen research articles, except method papers, describing the operating procedures for a particular model of equipment. We have added the website for the protocol in Table S1.

Lines 93-97: Please add more details about the procedure used to measure the specimens, associated errors and discuss the potential stress that this manipulation may represent and the effect it could have on the final results

RESPONSE: We have now rewritten the procedure and the discuss the potential stress in section 2.3.

The order of paragraphs in the Material and methods section is, at present, a bit confusing and should follow a more logical pattern. I would suggest that the paragraph on experimental design should show the setting, the replication, times etc., for all type of analyses. Then a paragraph on procedures for physiological analyses should explain all the used methods for respiration rates, survival rates, feeding rates and shell growth. Then a final paragraph on the shell composition should follow.

RESPONSE: We appreciate reviewer's suggestions and have changed the order accordingly.

Lines 99-100: Why is the color of new shell very different from the ancient one? Is it normal? Maybe you should discuss it somewhere. One would think that it is a culturing effect on shell structure...
Figure S1 should be in the main text.

RESPONSE: The colour of new shells should be white due to the presence of calcium carbonate. Yet, the colour will turn slightly yellowish over time because of the biofilm (e.g. bacteria, algae, etc.) growing on the surface (Fig. 1). It is very normal and important because the biofilm allows polychaete larvae to settle and form a colony. We have now put Figure S1 in the main text as Fig. 1.

Lines 103-112 should be part of the "Experimental set up" paragraph

RESPONSE: We have changed the order of paragraphs as suggested above.

Line 123: The specimens used for organic matter composition are the same measured for toughness? Please detail this kind of information

RESPONSE: It cannot be, unfortunately, because removing the organic matter in the shell at 550°C can substantially affect shell toughness. We have now clarified this (Ln 170-171).

Paragraph 123-129: 1) for the composite shell power did you mix calcite from different specimens, isn't it? Did they come from a same replicate for a treatment or even replicates were mixed? How did you prepare the powder exactly? Did you wash the shell before to avoid contamination (from algae given as food for example)? How? Why did you choose Mg/Ca ratio as a parameter to be measured?

RESPONSE: For each composite sample, shells from 3-5 individuals in the same treatment were used (Ln 173-174) and we had 3 composite samples as replicates per treatment for the geochemical properties. The powder was prepared by removing the newly-produced shells, rinsing them with deionized water (to remove the microalgae and other debris), drying them at room temperature and finally grinding them

using a mortar and pestle (Ln 175-177). Many calcifying organisms can change Mg/Ca in their calcitic shells in response to the changing environment (e.g. ocean acidification). The underlying mechanism remains largely unknown, but may be associated with metabolic energy that can be greatly affected by hypoxia. Therefore, we expected that Mg/Ca would change in response to hypoxia.

Paragraph 148-155: How did you calibrate oxygen probes before the analysis?
RESPONSE: Before the analysis, the probe was calibrated by inputting the value of calibration slope in the software. Then, we have to validate the DO concentration using another DO meter (e.g. handheld DO meter), which is calibrated by measuring the DO concentration of oxygen-saturated seawater. We may need to change the value of calibration slope until the DO concentrations between DO probes are the same. We have added the website for the operation manual of the oxygen probe in Table A1.

Line 152: "The air inside the syringe...": what air? Why is it there during the first measure? This part is not clear to me.
RESPONSE: The air is atmospheric air (Ln 140). For the initial measurement, the individuals were allowed to rest in the syringe for 15 min and the small pocket of air can help buffer the change in DO concentration during this period. We have added this information for clarity (Ln 140-141).

Line 154: "by gently stirring the FSW inside the syringe". How did you avoid oxygenation at this step?
RESPONSE: Such gently stirring can only help homogenize the DO concentration in the water body. Oxygenation of water is caused by dissolving atmospheric oxygen in water. However, this process cannot occur because atmospheric air cannot interact with the water inside the syringe, which is under air-tight conditions (Ln 141).

Line 156: Please specify the unit used for consumed oxygen. Normally mL or μmol are used. In your figure 4a you use μg $O_2$, which is quite unusual as a unit for oxygen.
RESPONSE: For the unit of consumed oxygen, mg $O_2$ is frequently used, while μg $O_2$ is just a conversion for the magnitude to avoid many decimal places (Ln 144-145).

Lines 159-160: Why did you only used one algal species for this experiment? Can you add fundamental details such as if the experiment was performed under light conditions, please? Also you say that 5 replicated bottles were used per treatment. In the first materials and methods section you say you have 3 replicated bottles with 10 specimens for the experiments. I'm lost... Are these different bottles? How many individuals per bottle per treatment do you have then?
Statistical paragraph: Where data transformed before the analyses to homogenize the magnitudes? How many permutations were performed? Which distance parameter was used and why? Can you specify the "aforementioned parameters" of line 168, please?
RESPONSE: One algal species should be used for the feeding experiment to avoid the selective feeding of filter feeders (due to different sizes and/or textures between microalgae), which can complicate the calculation and interpretation. We have now stated that the experiment was performed under light conditions (Ln 151). The experimental design (e.g. number of replicates) has been revised in a more explicit way in the Materials and Methods section. For univariate analysis, data are on the same scale (i.e. carrying the same unit), therefore transformation is unnecessary. The number of permutation is 999 and Euclidean distance was used which is commonly applied (Ln 199-200). We have listed the parameters (e.g. respiration rate, clearance rate, shell toughness, etc.) in the revision (Ln 200-202).

Line 172: You say hypoxia slightly hindered, but your statistics say this difference is significant, so maybe this should be emphasized a bit more.

RESPONSE: Please note that "statistically significant" does not necessarily imply "biologically significant". We can emphasize the statistical significance a bit more (Ln 205), but it is more important to compare the two factors with respect to the magnitude of change, which is of biological importance.

Line 174: Please replace "negligible" with "no".
RESPONSE: "no" is too absolute. We prefer to be conservative by using "no significant" (Ln 207).

Line 180: You say "but only slightly by non-lethal shell damage". It looks like a significant difference, visually (fig. 4a). Is it confirmed by statistical analyses? Yes, so you should say it!
RESPONSE: It is statistically significant (Ln 214).

Lines 181-182: specify whether statistical differences are visible and where.
RESPONSE: We have specified it (Ln 215).

Line 187: what do you mean by "ramifications"?
RESPONSE: It can mean changes in species populations, community structure and ecosystem functioning (Ln 220-221).

Line 188: "tolerant to hypoxia", at what temporal scale?
RESPONSE: Short-term scale (Ln 222).

Line 194: I would suggest not to use "unthreatened conditions" as a general for "shell damage", because hypoxia also is an unthreatened condition, so it can result confusing.
RESPONSE: We appreciate this suggestion, but we have explicitly defined "unthreatened conditions" as "individuals without shell damage" in the methods and mentioned the definition again in this sentence. There should be no confusion.

Line 195: "hypoxia slightly hinders"... again, is it significant or not?
RESPONSE: Significant (see Table S2).

Lines 201-202: redundant concept, already said.
RESPONSE: We have deleted this sentence.

Lines 205-206: Please replace "shell growth" with "inorganic components of the shell". You suggest that under hypoxia the production of organic matter compensate for diminished quality of inorganic components (>ACC). Palmer (1992) on the contrary suggests that organic matter production is costly and that would be the reason why high-organic calcareous microstructures became rare with evolution. How do you explain this incoherence?
RESPONSE: Suggestion for the word choice have been adopted (Ln 240-241). However, we never suggest that "under hypoxia the production of organic matter compensate for diminished quality of inorganic components (>ACC)". In this paragraph, we discussed the effects of hypoxia on shell growth and shell strength under unthreatened conditions. As the organic matter content was not affected by hypoxia, we suggest that the polychaetes still allocate similar amount of energy to maintain shell strength at the expense of shell growth under hypoxia.

Line 211-212: Could the overproduction of organic matter be related to higher calcification rates?

RESPONSE: Although we cannot rule out this possibility, we think it is still premature to make this speculation because we found that organic matter content does not necessarily increase with calcification rate (Normoxia vs. Hypoxia without shell damage).

Lines 219-221: I do not understand what you mean
RESPONSE: This sentence means when the individual is under life-threatening conditions and chance of survival becomes very low, it has to prioritize defence response as the last resort to maximize survival rate. We have rephrased this sentence to better illustrate the idea (Ln 255-257).

Line 224: You say the effect of hypoxia on defense response is not discernible. Although toughness is not affected, I would say that reduced shell growth rates (visible in figure 1) should be taken into account, to discern the effects of hypoxia, as well.
RESPONSE: We have revised this sentence by considering the effect of hypoxia on shell growth (Ln 260-261).

Line 234: please replace "signifies" with "may suggest".
RESPONSE: Suggestion adopted (Ln 271).

Line 241: At the end of the sentence you should add ".. can generally be maintained at least under slight hypoxia, on a short timescale."
RESPONSE: Suggestion adopted (Ln 278).

Line 248: please delete "and therefore *H. diramphus* prioritized defense response".
RESPONSE: Suggestion adopted.

Line 259: please add "open" between marine and waters (because "costal" are also marine waters!)
RESPONSE: Suggestion adopted (Ln 294).

Line 264: when you say that the defense response can be sustained you should specify "on a short timescale", because your results are based on short-term experiments.
RESPONSE: We appreciate this suggestion and have specified this in the text (Ln 299).

Lines 330-339: Please order the references on a chronological basis.
RESPONSE: Suggestion adopted (Ln 372-384).

Line 338 and 368: add authors to the list
RESPONSE: Authors added (Ln 403, 416).

Survival rate figure (S2) should not be in supplementary material, as this is an important result to take into account in the analysis of all the others.
RESPONSE: We have now described the results in the text (Ln 118-120).

**Reviewer 2**

The manuscript by Leung and Cheung provides information regarding how calcification processes in polycheate worms could be influenced by future hypoxia. The results are pretty straight forward, and I consider these types of studies are important, although not ground-breaking. I have a few concerns that should be addressed before this manuscript could be accepted.

RESPONSE: We are pleased to see that the reviewer recognizes the importance of this work.

The grammar and style should be improved in the introduction and discussion of this m/s before it could be published in any outlet. There are too many examples of this for me to highlight every one, but for example the use of the term "defence response". I suggest the authors ask a senior colleague who is a native English speaker to read over and correct for them. In general there is also a lot of speculation for a 21 day long study.

RESPONSE: Being senior English writers, we believe that the overall quality of English writing is good enough for publication based on our experience. Yet, we have polished the writing in the revision to maximize readability. The term "defence response" is widely used in the literature to describe the defensive behaviours of organisms at both individual (e.g. anti-predator response) and tissue/cellular (e.g. immune response) levels.

Specific comments: Line 40: In general I agree that calcification costs energy, but in some organisms the energy-dependence has been postulated as low (e.g. in corals – see McCulloch et al. (2012)). So this may be true for gastropods, but not necessarily so for some other organisms. So this sentence needs to be balanced somewhat.

RESPONSE: Please note that McCulloch et al. (2012) examined the effect of ocean acidification, where the "low energy cost" only refers to the additional energy for pH regulation at the calcification site, while the energy cost of calcification (i.e. production of calcified structures) is not taken into consideration. Thus, this "low energy cost" is irrelevant to our study because hypoxia does not affect the acid-base balance of organisms.

Line 70: The hypotheses around phenotypic plasticity needs to be strengthened and clarified. What exactly is the phenotype that is plastic here? The capacity to form different types of mineral in the shell? Or simply that responses will differ between control and reduced O2 concentrations? Reading the discussion, I think the authors are misusing the term phenotypic plasticity. Demonstrating variability in responses of individuals within the same population to a stressor is not demonstrating phenotypic plasticity, nor is demonstrating a different response under different treatments between different individuals.

RESPONSE: In ecology, "phenotypic plasticity" means that individuals can change their phenotypic traits (e.g., growth, behaviour, shell properties, etc.) in response to altered environmental and biological conditions, which has adaptive values (Malausa et al., 2005). In this study, the mineralogical properties of shells were proven to be plastic phenotypic traits as they were modified in response to hypoxia. The term "phenotypic plasticity" may be a bit general because the phenotypic traits in this study were only associated with mineralogical properties. To be more explicit, we will replace "phenotypic plasticity" with "mineralogical plasticity" (Leung et al., 2017) or similar wordings in the revision.

Leung, J.Y.S., Russell, B.D., and Connell, S.D.: Mineralogical plasticity acts as a compensatory mechanism to the impacts of ocean acidification. Environ. Sci. Technol., 51, 2652–2659, 2017.

Malausa, T., Guillemaud, T., and Lapchin, L.: Combining genetic variation and phenotypic plasticity in tradeoff modelling. Oikos, 110, 330–338, 2005.

Line 82: How was pH measured, and on what scale, using what buffers? More information needed here. How was salinity and temperature measured? I see some of these details in table S1, but there are required in the methods section.

RESPONSE: The pH was measured on NBS scale by a pH meter, calibrated using NBS buffers (Table S1). Temperature and salinity were measured using a thermometer and refractometer, respectively (Table S1). Since they are not the key parameters in this study, it is better to keep them in the Supplementary Information.

Statistical analysis: why use a permanova? I would expect each parameter to be separated analysed using univariate analyses as a first step. A justification for using permanova over an anova or linear model needs to be justified here.

RESPONSE: Please note that PERMANOVA can be used for univariate analysis, as applied in our study. With the use of Euclidean distance matrix, PERMANOVA can produce the same F-statistics as traditional ANOVA (Anderson, 2001).

Anderson, M.J.: A new method for non-parametric multivariate analysis of variance. Aust. Ecol., 26, 32–46, 2001.

Line 191-192: But this statement is at odds with the findings of the permanova and the figures, that calcification was impacted by hypoxia in this study. Also, the end of the sentence that this could be due to phenotypic plasticity needs to be explained, as this makes no sense to me.

RESPONSE: At this point of discussion ($1^{st}$ paragraph), we only make a general summary of the findings, which can help readers quickly grasp the key message of this study. The detailed explanation for each finding is provided in the subsequent paragraphs. We have revised the statement about the impact of hypoxia on calcification for clarity (Ln 224-226).

References used in the review: McCulloch MT, Falter J, Trotter J, Montagna P (2012) Coral resilience to ocean acidification and global warming through pH up-regulation. Nature Climate Change, 2, 623-627.

**Reviewer 3**

Dear Editor,

The manuscript "Calcification and inducible defense response of a calcifying organism could be maintained under hypoxia through phenotypic plasticity" by Leung and Cheung raises a few interesting questions regarding organismal adaptations to hypoxia. Although authors present a few interesting ideas, I do not find the manuscript in its current form suitable for publication due to a significant shortfall in methods description, exaggerated interpretation of results and an incoherent discussion. However, if the Discussion paper manuscript progresses for publication in Biogeosciences, I suggest that the authors consider some major revisions that I have described in detail in the supplement. Please find attached my suggestions for major and minor revisions.

RESPONSE: We thank this reviewer for reviewing and giving suggestions for our manuscript. We found that most of the comments are related to the methods and can be clarified easily. The only two comments for the discussion section are minor and can be easily addressed as well. Therefore, we are happy to incorporate the good suggestions into the revision.

Major Comments:

Title: I do not find this title representative of the authors results/discussion. Please describe which aspects of the phenotype are considered plastic, since there is no change in mechanical strength and authors discuss less crystallographic control and magnesium regulation under hypoxic conditions.

RESPONSE: The plastic traits are the mineralogical properties of shells. We have now made the title more descriptive (Ln 1-2).

Lines 34-38: Authors seem to go around a point here. Please state exactly what drives calcification instead of pH and seawater chemistry.

RESPONSE: The proposed main driver for calcification is the energetics of organisms (Ln 40).

Line 54: Energy costs can increase rather than reduce, if organic content is modified such that it increases.

RESPONSE: We have revised the sentence for clarity (Ln 56-57).

Line 76: Mean size of polychaetes?

RESPONSE: The tube length of polychaetes mostly ranged from 35 to 45 mm (Ln 78).

Line 102: Diameter of hole drilled?

RESPONSE: About 2 mm (Ln 105).

Line 103: Tubes were glued using what?

RESPONSE: Hot-melt adhesive (Ln 107).

Line 114: How was the shell fragment acquired and cleaned of organic tissue?

RESPONSE: Shell fragments were obtained by breaking the newly-produced shells using a pair of forceps and they were then cleaned by rinsing with deionized water (Ln 160-161). Please note that the flesh of polychaetes does not attach to the shell.

Line 116: Was the same surface (e.g: inner shell surface) always used for indentation?

RESPONSE: Yes, we always used the inner shell surface for indentation (Ln 164).

Line 122: Organic content of shell or whole polychaete? It is of the shell I assume, but is unclear.

RESPONSE: Only the newly-produced shells were used for analysis of organic matter content (Ln 170).

Line 126: How was the shell powder acquired?

RESPONSE: Shell powder was obtained by grinding the newly-produced shells using a mortar and pestle (Ln 175-177).

Line 130: Please provide information on how these polymorphs are typically distributed in the organism. Are the polymorphs specific to the outer/inner layer of the shell?

RESPONSE: We do not have information on the distribution of these carbonate polymorphs. Yet, it is not necessary to know that because it is irrelevant to our research question.

Line 139: Were ACC, aragonite and calcite standards measured? Please explain why, if not.

RESPONSE: Standards are used for determination of absolute quantity, but only relative quantity of these parameters is needed in our study. Specifically, since relative ACC content is indicated by the peak ratio in the IR spectrum, it is not necessary to measure the standard as long as background calibration for the baseline is made (Chan et al., 2012; Leung et al., 2017). As for the calcite to aragonite ratio, we apply the calibration equation in a method paper (Kontoyannis and Vagenas, 2000), which is derived by using pure calcite and aragonite.

Chan, V.B.S., Li, C., Lane, A.C., Wang, Y., Lu, X., Shih, K., Zhang, T. and Thiyagarajan, V: $CO_2$-driven ocean acidification alters and weakens integrity of the calcareous tubes produced by the serpulid tubeworm, *Hydroides elegans*. PLoS ONE, 7, e42718, 2012.

Leung, J.Y.S., Connell, S.D., Nagelkerken, I. and Russell, B.D.: Impacts of near-future ocean acidification and warming on the shell mechanical and geochemical properties of gastropods from intertidal to subtidal zones. Environ. Sci. Technol., 51, 12097–12103, 2017.

Kontoyannis, C.G. and Vagenas, N.V.: Calcium carbonate phase analysis using XRD and FT-Raman spectroscopy. Analyst, 125, 251−255, 2000.

Line 140: Diameter of the KBR-shell powder disc?

RESPONSE: 13 mm (Ln 193).

FTIR: FTIR is a bulk measurement and ideally should not be used to infer "relative" proportions of carbonate polymorphs. Typically, the presence of a 713 cm-1 peak is indicative of crystalline calcium carbonate comprising the bulk of shell carbonates. However, I am aware that this interpretation has been used before and if authors proceed with the analyses, could they please clarify if the spectra were scaled so that 713 cm-1 peaks had the same heights as described in Weiss et al (2002)? In addition, please specify the typical size of crystallites in shell since such ratios have been demonstrated to be influenced by particle size (Kristova et al 2015).

RESPONSE: FTIR has been widely used to indicate the relative ACC content by measuring the peak ratio between 856 cm$^{-1}$ and 713 cm$^{-1}$ (e.g. Beniash et al., 1997; Chan et al., 2012; Leung et al., 2017). Since relative ACC content is indicated by this ratio rather than an absolute peak height, it is not necessary to rescale the peak height at 713 cm$^{-1}$ to that in Weiss et al. (2002). The particle size of shell powder was ~5 µm (Ln 177).

Beniash, E., Aizenberg, J., Addadi, L., and Weiner, S.: Amorphous calcium carbonate transforms into calcite during sea urchin larval spicule growth. Proc. R. Soc. B, 264, 461–465, 1997.

Line 148: What were the syringes made to of?

RESPONSE: Polypropylene plastic (Ln 135).

Line 156: Hunger is only standardised if individuals were at the same start point.

RESPONSE: Therefore, all individuals were starved for 1 day prior to feeding trials. This is more than enough for them to clear their gut content.

Lines156-166: This doesn't represent clearance rates during the experiment.

RESPONSE: We disagree. This clearance method has been widely applied for determination of clearance/filtering/feeding rate of feeding feeders (e.g. Riisgård, 2001; Contreras et al., 2012; Leung et al., 2013; Leung and Cheung, 2017) (Ln 147-148).

Contreras, A.M., Marsden, I.D., and Munro, M.H.G.: Effects of short-term exposure to paralytic shellfish toxins on clearance rates and toxin uptake in five species of New Zealand bivalve. Mar. Freshw. Res., 63, 166–174, 2012.

Leung, J.Y.S. and Cheung, N.K.M.: Feeding behaviour of a serpulid polychaete: Turning a nuisance species into a natural resource to counter algal blooms? Mar. Pollut. Bull., 115, 379–382, 2017.

Leung, Y.S., Shin, P.K.S., Qiu, J.W., Ang, P.O., Chiu, J.M.Y., Thiyagarajan, V., and Cheung, S.G.: Physiological and behavioural responses of different life stages of a serpulid polychaete to hypoxia. Mar. Ecol. Prog. Ser., 477, 135–145, 2013.

Riisgård, H.U.: On measurement of filtration rates in bivalves — the stony road to reliable data: review and interpretation. Mar. Ecol. Prog. Ser., 211, 275–291, 2001.

Line 170-180: Please provide full FTIR spectra as a supplementary figure.

RESPONSE: We have now provided FTIR spectra in Fig. A2.

Lines 267-268: Can inferences be made regarding whether inner/outer layers were calcified if the polymorphs are specific to a layer of the polychaete shell?

RESPONSE: We cannot make this inference based on our results. As mentioned above, further investigation on structural properties is needed to answer this question, which is beyond the scope of this study.

Line 233-234: This is a strong statement. Regulation of Mg may be interpreted but the authors results do not signify that it is relaxed under hypoxia.

RESPONSE: We have toned down this statement (Ln 271-272).

Line 256-257: Please delete this final sentence. It is a very strong statement and the whole paragraph does not explain why hypoxia the key stressor in the future (which is debatable anyway).

RESPONSE: We have deleted this sentence.

Minor Comments:

Line 10-11: Sentence like this needs a reference.

RESPONSE: In the Abstract, citations should be avoided.

Line 25: change "shells" to skeletons.

RESPONSE: We have replaced "shells" with "shells or skeletons" (Ln 26).

Line 32: Delete "however".

RESPONSE: Suggestion adopted.

Line 368: Please provide full reference.

RESPONSE: Suggestion adopted (Ln 403).

Figure 5 (SEM): Are these images of the aragonite or calcitic parts of the shell? The legend needs more descriptive text. It is not obvious to me how these images indicate shell integrity.

RESPONSE: We cannot identify the type of carbonate mineral based on these images and this is beyond the scope of this imaging analysis. We have elaborated the figure legend to indicate shell integrity in terms of crystal thickness and density.

Table A1: Please include other calculated parameters such as HCO3-, CO32- and CT.

RESPONSE: Suggestion adopted (Table A1).

References used for review:

Weiss et al (2002) Mollusc larval shell formation: amorphous calcium carbonate is a precursor phase for aragonite. DOI: 10.1002/jez.90004

Kristova et al (2015) The effect of the particle size on the fundamental vibrations of the [CO3(2-)] anion in calcite. DOI: 10.1021/acs.jpca.5b02942.

[revised manuscript text omitted]

Within Hypoxia: N.S. |
| DO × Context | 1 | 57.2 | 95.5 | **0.001** | |

---

## Author Response (AR2)

**SOUTHERN SEAS ECOLOGY LABORATORIES**
SCHOOL OF BIOLOGICAL SCIENCES
UNIVERSITY OF ADELAIDE, SA 5005, AUSTRALIA

21 April 2018

Dear Prof. Treude,

Please consider our REVISED manuscript entitled "Effects of hypoxia and non-lethal shell damage on shell mechanical and geochemical properties of a calcifying polychaete" for publication as a research article in *Biogeosciences*.

We appreciate the anonymous reviewers for their suggestions to further improve the impact of our manuscript. The changes have been stated in the Response with the line number and shown in the highlighted version of revised manuscript. Thank you very much for considering our manuscript again and we look forward to your positive reply at your earliest convenience.

Yours sincerely,

Dr. Jonathan Leung
Corresponding author
On behalf of Napo Cheung

Reviewer 1

This manuscript presents clear results from an interesting experiment investigating he impacts of hypoxia and shell damage on a species of calcifying polychaete. The results suggest calcification can be maintained after shell damage and during exposure to 3 weeks of hypoxia, however mechanical and physical properties change with exposure to shell damage and hypoxia.

RESPONSE: We are pleased to see that the reviewer found this study interesting.

Introduction

This was generally well written with a clear scientific question and means to answer it. I feel the concept of changing shell mineralogy should be introduced in more detail with clearer rational behind the expected outcomes in the experiment. Introducing the different energetic costs of aragonite/calcite production in polychaetes would make the scientific significance more robust.

RESPONSE: We agree that the concept of changing shell mineralogical should be introduced more, especially concerning the energy cost of calcification. We have now introduced how the energy cost of calcification could be altered by changing calcite/aragonite, Mg/Ca and ACC content (Ln 57-68), which can further clarify the rationale of our hypotheses.

Methods

I have a few minor problems with the methodology section. Oxygen consumption measurements were generally well explained however I do not understand the sentence "the final dissolved oxygen concentration of FSW was recorded when it becomes steady by gently stirring the FSW inside the syringe" (L142).

RESPONSE: The dissolved oxygen concentration of FSW at the bottom of the syringe was lower than that on the top due to the respiration of polychaetes (i.e. uneven dissolved oxygen concentration in the water column). To obtain an accurate reading, gently stirring the FSW inside the syringe is needed to ensure uniform dissolved oxygen concentration of the FSW. This reason has been added (Ln 153).

Additionally, a clear justification regarding the exclusive use of non-parametric tests for all the data should be provided.

RESPONSE: Please note that PERMANOVA is not a non-parametric test. While it is regarded as a semi-parametric test, it has been widely used as a more robust substitute for the traditional ANOVA because it is distribution-free (i.e. no need to meet the normality assumption) and it has the same statistical power as the traditional ANOVA with the same F-statistics. We have added an appropriate reference (Anderson, 2001) for those who feel interested in the theory about PERMANOVA.

Results

Slightly more detail could be given in the written description of the results however the figures and tables are excellent and all relevant data is presented.

RESPONSE: We are pleased to have reviewer's commendation for the presentation of data and we have described the results slightly more in the text as requested.

Discussion

This section was generally excellent however there are some patterns in the results which should be better discussed. The authors state the energy demand for calcification is enormous (L227). In relative terms compared to organic tissue production, calcification is a minute cost being ~ 15 x lower than the cost of organic tissue synthesis (Palmer 1983, Marine Biology). This idea should also be discussed further as in the non-lethal shell damage treatment, calcified structures contained more organic material than in the non damage treatment. This would suggest and even higher cost of shell production.

RESPONSE: This is a good suggestion. We have now provided further information that the enormous energy demand for calcification is mainly due to the production of organic matrix (Ln 241). In addition, we have mentioned that the polychaetes can invest a substantial amount of energy not only for shell growth, but also for production of energy-costly organic matrix to enhance shell mechanical strength in the life-threatening situation (Ln 271-274). This can clearly show that defence response is prioritized following non-lethal shell damage, regardless of the high energy cost of calcification.

I also feel that the changes in clearance rates are extremely important as the energy income dictates the magnitude of energy partitioning to different growth and metabolic processes. It is interesting that in the normoxic treatment, calcification rate was ~ 4 x higher in the shell damage treatment than control despite a ~ 3 x decrease in food cosumption compared to the control! This is quite contradictory and should be discussed with relation also to somatic tissue growth and maybe reproduction.

RESPONSE: We have mentioned this unexpected result in the discussion and agree with the reviewer that factors other than mineralogical properties may also cause this unexpected result. While we have now suggested that energy reserves may be used to support the increased shell growth (Ln 299-302), it is still premature to speculate too much without solid evidence of somatic growth and reproduction.

Lastly I think the discussion would benefit from a slight expansion of the section discussing changes in shell mineralogy. The authors correctly state that producing calcite shells is more energetically favourable than aragonite shells, however what is the magnitude of this change? If it is in the region of half the cost then it is indeed a significant change, however I don't believe that changing shell mineralogy will have that much of a significant effect on the energetic cost of shell production. Rather changing mineralogy will have more of an impact on dissolution capacity and shell strength.

RESPONSE: Based on the literature, producing calcite is less energy-costly than producing aragonite, but the magnitude is unfortunately not reported probably because the energy cost of calcification also depends on other factors, especially organic matter content. Given the changes in calcite/aragonite, Mg/Ca, ACC content and total organic matter content, we found that the polychaetes tend to minimize the cost of calcification under hypoxia. We have slightly expanded the

discussion by adding relevant examples to show that precipitation of calcite is boosted under metabolic stress conditions (Ln 284-287). The trade-offs of shell growth against shell quality have been discussed (Ln 305-308).

Additionally, there are also some grammatical mistakes at several occurrences in the manuscript which need correcting (L96, L244, L276, L294 among others.)

RESPONSE: We have polished the manuscript carefully throughout and believe that the English writing can meet the publication standard based on our experience.

RESPONSE: We have already read this paper and recognized that particle size can affect the peak ratio in the FTIR spectrum. Nevertheless, the particle size of shell powder used in our study was consistent (i.e. no bias) between treatment groups, meaning that the difference in the peak ratio across treatments was not caused by particle size. The particle size of shell powder used was already provided (Ln 187).

Line 107: Please provide company of hot melt adhesives.

RESPONSE: The company name (i.e. 3M) has now been added (Ln 117).

Please provide units (or specify if they are arbitrary) for figure A2.

RESPONSE: The unit is arbitrary. We have now added "(a.u.)" for Figure A2.

Please write CT correctly in Table A1 where C is italicized, and T is subscript. Further, the fourth line in the legend should read as "Temperature and salinity were measured daily…" The next sentence should also be corrected to "measured weekly".

RESPONSE: All suggestions for Table A1 have been adopted.

[revised manuscript text omitted]

Within Hypoxia: N.S. |
| DO × Context | 1 | 57.2 | 95.5 | **0.001** | |

---

## Author Response (AR3)

Editor's comments:

I am delighted to inform you that your manuscript has been accepted for publication in Biogeosciences. I would just like to ask you to replace the html links in Table A1 and replace it by proper (i.e., permanent) methodological citation.

RESPONSE: We are glad to see the acceptance of this manuscript. Thank you for handling the manuscript. We have replaced the html links in Table A1 by in-text citations and added the corresponding references (see supplementary information).